# ThinkBench: Dynamic Out-of-Distribution Evaluation for Robust LLM Reasoning

**Shulin Huang[1,2], Linyi Yang[3],[*]Yan Song[3], Shuang Chen[1], Leyang Cui[2], Ziyu Wan[4],**
**Qingcheng Zeng[5], Ying Wen[4], Kun Shao[6], Weinan Zhang[4], Jun Wang[3], Yue Zhang[2][*]**
[1]Zhejiang University, [2]School of Engineering, Westlake University,
[3]University College London, [4]Shanghai Jiao Tong University,
[5]Northwestern University, [6]Huawei Noah's Ark Lab
huangshulin@westlake.edu.cn

## Abstract

Evaluating large language models (LLMs) poses significant challenges, particularly due to issues of data contamination and the leakage of correct answers. To address these challenges, we introduce *ThinkBench*, a novel evaluation framework designed to evaluate LLMs' reasoning capability robustly. ThinkBench proposes a dynamic data generation method for constructing out-of-distribution (OOD) datasets and offers an OOD dataset that contains 2,912 samples drawn from reasoning tasks. ThinkBench unifies the evaluation of reasoning models and non-reasoning models. We evaluate 16 LLMs and 4 PRMs under identical experimental conditions and show that most of the LLMs' performance are far from robust and they face a certain level of data leakage. By dynamically generating OOD datasets, ThinkBench effectively provides a reliable evaluation of LLMs and reduces the impact of data contamination. Our data and codes are available at `https://github.com/huangshulin123/ThinkBench`.

## 1 Introduction

Large Language Models (LLMs) have made significant advancements across diverse domains, including language understanding [Karanikolas et al., 2023, Xu et al., 2024], language generation [Wu et al., 2024b, Naveed et al., 2023], and complex reasoning [Hao et al., 2023, Lu et al., 2023, Azerbayev et al., Wang et al., 2024a]. Reasoning models, such as o1 [OpenAI, 2024], have further extended the capabilities by leveraging the Process Reward Model (PRM) and advanced search strategies during decoding. Notably, models, such as OpenAI o1, o3 [OpenAI, 2024, 2025] and Deepseek-R1 [Guo et al., 2025], have shown performance that rivals that of a Ph.D.-educated individual, outperforming GPT-4 in complex reasoning tasks, revealing substantial potential for future developments in LLMs.

An interesting finding from the OpenAI report [OpenAI, 2024] is that the performance of the o1-series models on the Advanced International Mathematics Exam (AIME) significantly declined in 2024, compared to previous years (1983-2023: 0.74 vs. 2024: 0.50). Historical statistics, however, indicate that the median scores of human participants on the AIME exam in previous years are consistent with 2024 exam (4.81 vs. 5.0) [2]. Since o1 was trained on data by 2023, before the AIME 2024 Exam, this discrepancy suggests a possible data contamination issue, raising an important question about how to assess the generalization abilities of LLMs, rather than memorization.

To address this problem, we propose a novel robust evaluation framework, namely ThinkBench. Building on causal theory and semi-factual causality [Delaney et al., 2021, Kenny and Keane, 2021],

---

[*]Correspondence to: (yanglinyiucd@gmail.com, zhangyue@westlake.edu.cn)
[2]Sourced from https://artofproblemsolving.com.

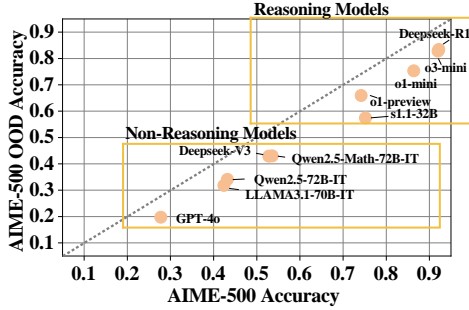 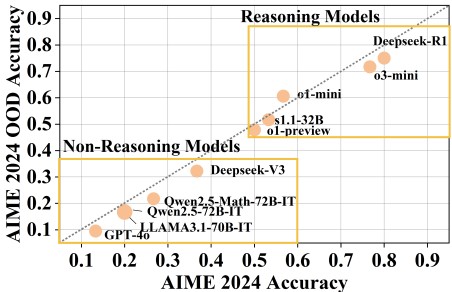

(a) OOD performance vs. ID performance for several reasoning models on AIME-500.

(b) OOD performance vs. ID performance for several reasoning models on AIME 2024.

Figure 1: **Math Reasoning Gap**: Most models demonstrate a visible performance gap between their math reasoning performance on ID and OOD, including open-source models and commercial models.

we introduce Out-Of-Distribution (OOD) data generation designed to test reasoning capabilities. We introduce scenario-level and attack-level semi-fact data generation methods, differing in the specific elements of text they alter, enabling the creation of evaluation datasets that are both robust and challenging. By decoupling reasoning from memorization, dynamic evaluation allows us to more effectively test how well LLMs can generalize to unseen reasoning scenarios.

We take AIME-500 (500 AIME questions from 1983 to 2023) and AIME 2024 (30 AIME questions in 2024) for math reasoning tasks, and GPQA Diamond for scientific questions, dynamically generating an OOD dataset of 2,912 samples, providing a diverse set of challenges that test both the generalization and reasoning capabilities of LLMs. As illustrated in Figure 1, compared to the original datasets, our OOD evaluation set proves to be more difficult, resulting in an average performance decay of 24.9% and 11.8% across all models on AIME-500, and AIME 2024, respectively. This indicates that there was indeed some data leakage in AIME questions before 2024, highlighting the importance of mitigating data contamination for reasoning evaluations. The difference in performance decay between AIME-500 and AIME 2024 demonstrates that our dynamically constructed OOD data construction is a convenient and effective method to reduce the impact of data contamination.

Figure 1 shows that o1 OpenAI [2024], o3 OpenAI [2025], Deepseek-R1 Guo et al. [2025] and s1 Muennighoff et al. [2025] maintain the strongest accuracy. As representatives of reasoning models Luo et al. [2025], Liu et al. [2025], o1, o3, and s1 enhance inference performance by increasing computational resources during testing, contrasting with non-reasoning models. ThinkBench provides a reasoning benchmark to evaluate both reasoning models and non-reasoning models.

In addition to the overall model accuracy, we also explore the impact of various PRMs and their performance under the best-of-*n* search during decoding. Fine-grained evaluations show how different data generation strategies, such as Math-shepherd [Wang et al., 2024c], influence model outcomes. The performance improves with an increased test-time computation budget, further highlighting the discriminative power of our benchmark and the quality of the data.

To our knowledge, we are the first to present a robust dynamic evaluation benchmark for testing reasoning capability in LLMs [Qin et al., 2024, Huang et al., 2024c, Wang et al., 2024a]. In ThinkBench, we provide a convenient and effective OOD data construction method and a high-quality dataset to reduce data contamination impact, effectively evaluating the reasoning ability in both reasoning models and non-reasoning models. Notably, we verify the validation of the test-time scaling law using the dynamic evaluation without data contamination based on ThinkBench.

## 2 Related Work

**Evaluating Large Language Models.** Evaluating LLMs accurately and fairly poses a significant challenge [Chang et al., 2024]. Mainstream evaluation methods include: **(1) LLMs-as-a-judge**: Benchmarks like AlpacaEval [Li et al., 2023], PandaLM [Wang et al., 2023b], MT-Bench [Zheng et al., 2023], and C-Eval [Huang et al., 2024b] use large language models for predefined ques-

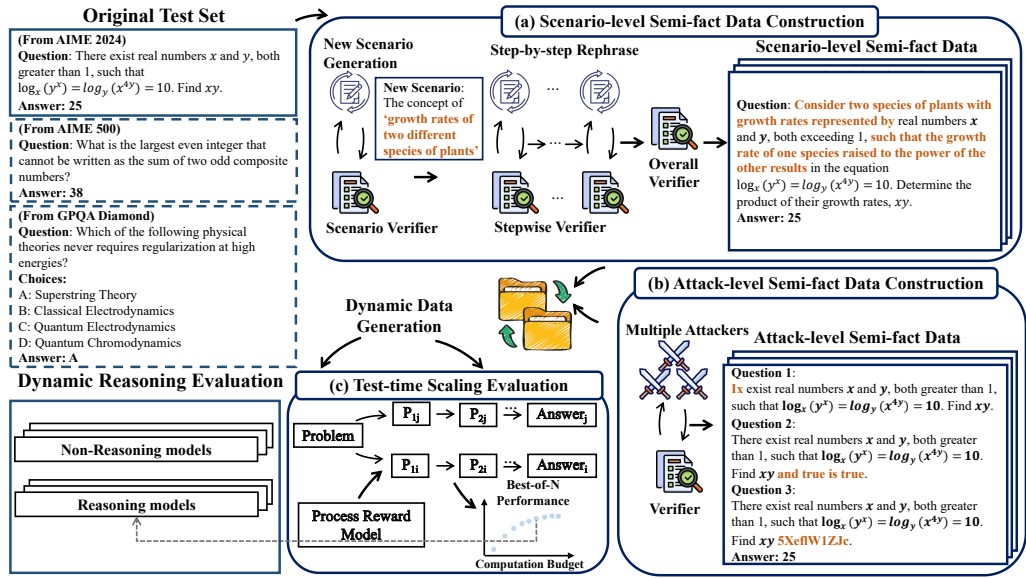

Figure 2: Overview of ThinkBench framework. Based on the original data, ThinkBench dynamically generates scenario-level Semi-fact Data (a) and Attack-level Semi-fact Data (b), which can be used to evaluate the robustness of reasoning models and non-reasoning models. ThinkBench can also serve as a useful tool for Test-time Scaling Evaluation(c).

tions. **(2) Humans-as-a-judge**: Human evaluation provides comprehensive feedback through expert reviews [Ribeiro and Lundberg, 2022, Gao et al., 2023]. **(3) Other benchmarks**: Several traditional benchmarks employ static datasets to assess models across various tasks [Liang et al., 2022, Hendrycks et al., 2020, 2021]. Our work falls into the third category. However, rather than using static data, we generate test sets dynamically.

**Robustness of Large Language Models.** Evaluating LLM robustness is crucial Muennighoff et al. [2025], Guo et al. [2025] across diverse applications Wang et al. [2023a], Glazer et al. [2024], Li et al. [2024b]. Prior work Li et al. [2024a] like OOD-GLUE [Yuan et al., 2023], GLUE-X [Yang et al., 2023], and ZebraLogic [Lin et al., 2024] focus on robustness of non-reasoning models. Specifically, ZebraLogic [Lin et al., 2024] emphasizes the generation of puzzles with controllable and quantifiable complexity. Yang et al. [2022] generate OOD data via semi-fact augmentation, while Hosseini et al. [2024] identify reasoning gaps in math problems and Wu et al. [2024a] introduce cofQA using counterfactual perturbations. Our work differs by specifically addressing reasoning tasks where OOD testing is more necessary as compared to general tasks. In addition, existing benchmarks predominantly emphasize non-reasoning models [Li et al., 2023, Hendrycks et al., 2020, Huang et al., 2024a, Li et al., 2024c]. For reasoning models, current evaluations often target specific tasks, such as planning [Wang et al., 2024b] and rule execution [Gui et al., 2024], with limited comprehensive robustness analyses [Zhong et al., 2024]. Unlike this line of work, our benchmark focuses on robustness and reasoning with practical applications, offering statistically significant insights.

## 3 Dynamic Evaluation Benchmark

Building on semi-factual causality from causal theory and XAI literature [Kenny et al., 2021, Kenny and Keane, 2021], we construct OOD data that alters surface-level problem presentation while preserving the underlying logical structure and correct answer to test whether models rely on genuine reasoning versus memorized patterns. Our core assumption is that robust reasoning models should maintain consistent performance when the same logical problem appears in different surface forms, as the required reasoning chain remains unchanged.

As shown in Figure 2, ThinkBench contains two semi-fact data generation methods, aiming to assess real reasoning ability of LLMs: (a) Scenario-level Semi-fact Data [Yang et al., 2023, Zhu et al., 2023a,

2024, Opedal et al., 2024], which changes the scenario for the original reasoning data; (b) Attack-level Semi-fact Data [Zhu et al., 2023b], which uses three attack methods to rephrase original data. To construct Out-of-Distribution data, we focus on variations in language, scenarios, and expressions while preserving the fundamental reasoning patterns to ensure fair evaluation of reasoning abilities.

We use generated OOD data to perform dynamic reasoning evaluation on both reasoning models and non-reasoning models. Maintaining core knowledge while altering scenarios or expressions, we evaluate whether models can consistently apply learned knowledge across contextual variations. Finally, we can leverage our OOD data to conduct (c) test-time evaluation based on PRMs.

## 3.1 OOD Data Generation

**Scenario-level Semi-fact Data Generation.** The process primarily involves two types of agents: the Rephrasing Agent, which is responsible for transforming the original questions and generating new ones, while the Verifier Agent, which is responsible for confirming at each step whether the rephrasing meets the current requirements and whether the rephrasing is valid.

There are two Rephrasing Agents. The first generates suitable scenarios for reasoning problems, ensuring that the new scenarios are appropriately transferable concerning the core of the original problem. The second rewrites each part (often at the sentence level) to fit the new scenario while preserving the original meaning. This step-by-step process uses each newly generated part as a reference for subsequent parts.

There are three Verifier Agents. The first checks if the core content of the original problem can be effectively transferred to the new scenario. The second evaluates each newly generated part, ensuring it meets three criteria: it conveys the same core meaning as the original, is consistent with previously revised parts, and is correctly adapted to the new scenario. Additionally, an overall Verifier Agent assesses the overall new problem, ensuring it maintains essential consistency with the original questions, preserving informational content, and is appropriately constructed within the new scenario.

These Verifier Agents implement our core assumptions by checking: (1) logical equivalence between original and rephrased problems, (2) answer correctness, (3) contextual coherence. Any rephrasing failing checks is rejected and regenerated, ensuring all OOD data satisfies our semi-factual criteria.

For the datasets that contain choices, similarly, Rephrasing Agent modifies the expressions of certain options and randomly rearranges their order. Rephrasing of choices must also pass the Verifier Agent's check to ensure the data is valid.

**Attack-level Semi-fact Data Generation.** We focus on realistic errors using three methods:

1. TextBugger (character-level) [Li et al., 2019]: This method simulates user input errors by introducing mistakes or typos within words.

2. CheckList (sentence-level) [Ribeiro et al., 2020]: This approach assesses model robustness by adding irrelevant or redundant sentences to the original text.

3. StressTest (sentence-level) [Naik et al., 2018]: This method evaluates model robustness by incorporating distracting content.

These methods reflect common errors such as typos and extraneous information. They serve as three Attack Agents. We apply only a single iteration of perturbation, avoiding unrealistic error densities. Attack-level Semi-fact data construction involves these three Attack Agents and a Verifier Agent. The Verifier Agent is used to check if it is consistent with the core aspects of the original data and ensures that any errors introduced by the perturbation do not impede overall comprehension.

For an original reasoning problem, we construct our OOD test by creating one scenario-level semi-fact data instance and three attack-level semi-fact data instances. The method for evaluating the model's OOD Accuracy for the original data $i$ is as follows:

$$\text{Acc(OOD)} = \frac{1}{2}\left(\min_{j=1}^{3}\text{Acc}(A_{ij}) + \text{Acc}(S_i)\right), \tag{1}$$

where $\text{Acc}(A_{ij})$ denotes the accuracy of performance for the $j$-th attack-level semi-data, with $j = 1, 2, 3$. $\text{Acc}(S)$ represents the accuracy of performance for the scenario-level semi-data $S$.

Table 1: Statistics of reconstructed reasoning datasets based on three original test datasets, including AIME-500, AIME 2024, and GPQA Diamond.

|  | AIME-500 | AIME 2024 | GPQA Diamond |
|---|---|---|---|
| # Samples of original | 500 | 30 | 198 |
| Questions' Avg Len | 51.1 | 60.1 | 67.7 |
| Choices' Avg Len | - | - | 27.8 |
| # Samples of OOD | 2,000 | 120 | 792 |
| Questions' Avg Len | 61.2 | 70.1 | 85.2 |
| Choices' Avg Len | - | - | 25.7 |

In conclusion, Scenario-level Rephrasing focuses on shifting the contextual scenario of a question while preserving the core knowledge and answer. In contrast, Attack-level Rephrasing modifies sentence structures (e.g., adding noise, redundancy) following prior adversarial methods [Li et al., 2019, Ribeiro et al., 2020, Naik et al., 2018] to mimic linguistic variations. This separation helps clarify which specific dimensions of rephrasing cause performance gaps, providing valuable insights for understanding model robustness across multiple facets of linguistic variation.

**Semi-fact Data Construction.** As shown in Figure 2, Scenario-level semi-fact data is constructed as follows: The original query, "There exist real numbers $x$ and $y$, both greater than 1, such that $\log_x(y^x) = \log_y(x^{4y}) = 10$. Find $xy$." is transformed step-by-step within a new scenario "The concepts of growth rates of two different species of plants" into "Consider two species of plants with growth rates represented by real numbers $x$ and $y$, both exceeding 1, such that the growth rate of one species raised to the power of the other results in the equation $\log_x(y^x) = \log_y(x^{4y}) = 10$. Determine the product of their growth rates, $xy$."The Verifier Agent then checks the rephrased question to ensure that the rephrased question is in a reasonable scenario and the rephrased expression does not alter the core topic of the question.

For Attack-level semi-fact data, TextBugger injects character-level noise (e.g., replacing "There" with "Ix"), while CheckList and StressTest append syntactically valid but irrelevant suffixes (e.g., "5XeflW1ZJc" and "true is true") to the problem statement. The Verifier Agent also needs to ensure the validity of this rephrased version.

## 3.2 Data Analysis

We construct three OOD semi-fact datasets from the following datasets to evaluate the model's reasoning ability: **AIME-500 (extracted from Year of 1983 - 2023) and AIME 2024 (Year of 2024).** The AIME dataset is designed to challenge the most talented high school mathematics students in the United States. The original test component of AIME-500 consists of 500 questions extracted from the original dataset [Zheng et al., 2021] while AIME 2024 contains 30 examples included in the exam for 2024. **GPQA Diamond.** Originating from the GPQA Diamond dataset [Rein et al., 2023] which tests scientific questions. These original datasets are in English, publicly available, and permitted for research. AIME [3] is CC0-licensed (Public Domain) and GPQA Diamond [4] uses the MIT License.

**Statistics Analysis.** Table 1 shows the analysis of three dynamically constructed datasets. For Out-Of-Distribution (OOD) test, the sample size is four times as the original due to each original instance being transformed into one Scenario-level and three Attack-level Semi-fact Data entries. This augmentation allows for a thorough OOD evaluation of model performance. The OOD test data exhibits a length comparable to that of the original test data, indicating that both Scenario-level and Attack-level Semi-fact data, derived through part-by-part or attack-based modifications respectively, preserve the model's comprehension of the question's core elements. This construction effectively incorporates legitimate semantic information that rephrases the question without altering its answer. This approach helps distinguish models that rely on memorization rather than reasoning by highlighting the differences between OOD and ID data.

**Human Evaluation.** To verify whether the dynamically constructed data, deemed legitimate by the model, also corresponds to the correct answers from a human perspective, we further design a

---

[3]Sourced from https://huggingface.co/datasets/gneubig/aime-1983-2024.
[4]Sourced from https://github.com/idavidrein/gpqa.

Table 2: Reasoning performance. We report performance on OOD data of different models on three datasets that we dynamically construct. Each performance value represents the accuracy in solving problems from the dataset. The accuracy of performance on OOD is calculated through Equation 1. The term "Gap" denotes the percentage decrease in performance on Semi-fact data tests compared to the original test performance.

| | AIME 2024 | | | AIME-500 | | | GPQA Diamond | | | AVG Gap (%) |
|---|---|---|---|---|---|---|---|---|---|---|
| | Original | OOD | Gap (%) | Original | OOD | Gap (%) | Original | OOD | Gap (%) | |
| o1-preview | 0.500 | 0.484 | -3.3 | 0.742 | 0.659 | -11.2 | 0.684 | 0.638 | -6.8 | -7.1 |
| o1-mini | 0.567 | 0.600 | 5.8 | 0.864 | 0.753 | -12.8 | 0.592 | 0.564 | -4.7 | -3.9 |
| o3-mini | 0.767 | 0.717 | -6.5 | 0.922 | 0.834 | -9.5 | 0.727 | 0.710 | -2.4 | -6.1 |
| Deepseek-R1 | 0.800 | 0.750 | -6.3 | 0.920 | 0.828 | -10.0 | 0.747 | 0.682 | -8.7 | -8.3 |
| GPT-4o | 0.133 | 0.100 | -24.8 | 0.278 | 0.197 | -29.1 | 0.495 | 0.447 | -9.8 | -21.2 |
| Deepseek-V3 | 0.367 | 0.333 | -9.3 | 0.528 | 0.429 | -18.8 | 0.581 | 0.530 | -8.8 | -12.3 |
| Mixtral-8x7B-IT-v0.1 | 0.000 | 0.000 | - | 0.012 | 0.006 | -50.0 | 0.168 | 0.163 | -3.0 | -17.7 |
| Qwen2.5-72B-IT | 0.200 | 0.184 | -8.3 | 0.432 | 0.341 | -21.1 | 0.536 | 0.487 | -9.1 | -12.8 |
| Qwen2.5-Math-72B-IT | 0.267 | 0.217 | -18.9 | 0.536 | 0.430 | -19.8 | 0.449 | 0.416 | -7.5 | -15.4 |
| LLAMA3.1-70B-IT | 0.200 | 0.167 | -16.5 | 0.424 | 0.318 | -25.0 | 0.388 | 0.383 | -1.3 | -14.3 |
| s1.1-32B | 0.533 | 0.517 | -3.0 | 0.752 | 0.574 | -23.7 | 0.525 | 0.508 | -3.3 | -10.0 |
| Gemma-2-27B-IT | 0.033 | 0.017 | -50.0 | 0.062 | 0.045 | -27.4 | 0.291 | 0.281 | -3.6 | -27.0 |
| Gemma-2-9B-IT | 0.000 | 0.000 | - | 0.032 | 0.021 | -34.4 | 0.214 | 0.189 | -11.9 | -15.4 |
| LLAMA3.1-8B-IT | 0.000 | 0.017 | - | 0.132 | 0.087 | -34.1 | 0.204 | 0.179 | -12.5 | -15.5 |
| Phi-3-mini-4k-IT | 0.000 | 0.000 | - | 0.046 | 0.026 | -43.5 | 0.224 | 0.192 | -14.5 | -19.3 |
| LLAMA3.2-3B-IT | 0.033 | 0.033 | 0.0 | 0.122 | 0.087 | -28.7 | 0.153 | 0.161 | 4.9 | -7.9 |

human evaluation. Specifically, we use all samples from AIME 2024 for evaluation, which has 30 samples. We provide a detailed explanation of the annotation guidelines to 3 undergraduate students proficient in mathematics and pay them $2.5 per entry for manual annotation. The result of the human evaluation shows that the questions and answers for the 30 corresponding Scenario-level Semi-fact data and Attack-level Semi-fact data were 100% aligned. This perfect alignment validates that our perturbations satisfy the invariance assumption, confirming that reasoning chains remain intact despite surface-level distributional shifts. To further validate the quality of the constructed data, we present the details of the further validation in Appendix A.5.

# 4 Experiments

We conduct experiments to verify the effectiveness of our proposed dynamic OOD data construction method and analyze the differentiated performance and robustness of various LLMs.

## 4.1 Setup

We use the datasets in Section 3.2 for our experiment across two test sets: (1) the original test set, (2) the OOD test set. We conduct evaluations on various models, including o1-preview [OpenAI, 2024], o1-mini [OpenAI, 2024], o3-mini [OpenAI, 2025], GPT-4o [Achiam et al., 2023], Deepseek-V3 [Liu et al., 2024], Deepseek-R1 [Guo et al., 2025], S1.1-32B [Muennighoff et al., 2025], LLAMA3.1-70B and 8B [Dubey et al., 2024], Gemma2 (both 9B and 27B) [Rivière et al., 2024], Mistral-7B [Jiang et al., 2023], Mixtral-8x7B [Jiang et al., 2024], Qwen2.5-72B-IT [Yang et al., 2024], and Qwen2.5-Math-72B-IT [Yang et al., 2024]. All models were configured with a temperature setting of 0.7 and used the pass@1 metric in a single test run. Furthermore, for the open-source models, each experiment was conducted on a computing resource under Linux OS, including 8 GPUs (NVIDIA H100 80GB HBM3) and 2 CPUs (Intel Xeon Platinum 8558 Processor).

## 4.2 Results

The overall results of ThinkBench are shown in Table 2. We observe that all models exhibit a certain degree of performance decline when evaluated on the original dataset versus the OOD dataset. This decay in performance can be attributed to the fact that the models have encountered, to some extent, the original dataset. Notably, the performance gap between ID (Original) and OOD in AIME 2024 for most models is much smaller than the performance gap in AIME-500, which demonstrates the data leakage in AIME data before 2024. The existing dataset [Li et al., 2024a, Glazer et al., 2024] is insufficient to reveal the aforementioned phenomenon.

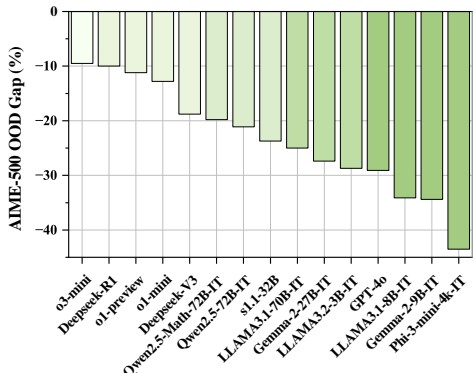 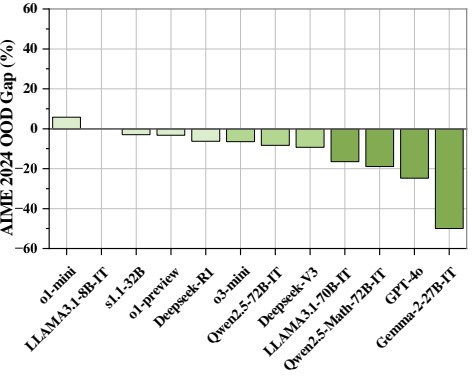

(a) OOD performance vs. ID performance for AIME-500.

(b) OOD performance vs. ID performance for AIME 2024.

Figure 3: The performance gap between ID and OOD test on AIME-500 and AIME 2024. "ID performance" and "OOD performance" represent the accuracy of LLMs in solving problems on the AIME-500 and AIME 2024's original test and OOD test, respectively.

We also observe that models with fewer parameters generally perform worse and are less robust than those with more parameters, especially within the Llama 3.1 model family. As the parameter count increases, both performance and robustness improve significantly, consistent with scaling laws. Thus, the number of parameters is crucial for ensuring robustness.

### 4.2.1 Math Reasoning

Figure 1 presents a comparative analysis of the performance of various LLMs on AIME-500 and AIME 2024, specifically contrasting their performance on the original ID data with that on OOD data. The results on the original set are mainly consistent with previous work Glazer et al. [2024], Li et al. [2024b]. Notably, the dashed line at a 45-degree angle from the origin signifies, where a robust model exhibits nearly equivalent performance on both OOD and ID data. These two datasets primarily evaluate mathematical reasoning and logical capabilities, featuring relatively complex problem types.

**AIME-500 vs. AIME 2024.** It is evident that o1 series models, o3, Deepseek-R1, and s1 exhibit a small decline in performance on the latest 2024 AIME. Moreover, the performance gap between OOD and ID is markedly larger on AIME-500 for nearly all models, including the o1 series, compared to the OOD and ID gap observed on AIME 2024. Specifically, the o1-preview shows a performance gap of -3.3% on AIME 2024 and -11.2% on AIME-500, while the o1-mini model exhibits a gap of +5.8% and -12.8%, respectively. The most advanced reasoning models – o3-mini, Deepseek-R1 and s1.1-32B – show the same phenomenon. This indicates that there was indeed some degree of data leakage in questions before 2024.

Ideally, a robust model that has genuinely learned to apply knowledge for reasoning should not exhibit significant performance degradation when the expression and scenario of a question change, provided that the core knowledge being assessed remains the same. Additionally, s1.1-32B shows a significant gap in AIME-500 ID-OOD performance. One possible explanation for its low robustness is the limited amount of data used for training. Furthermore, most models fall below the dashed line, indicating a decline in performance when transitioning to OOD data. This phenomenon reveals their limitations in generalizing to previously unseen complex mathematical problems.

**ID vs. OOD.** Figure 3b and Figure 3a illustrate the percentage difference in performance between OOD and ID for various models on the AIME 2024 and AIME-500. Certain models, such as o3-mini, Deepseek-R1, and o1-series models distinguish themselves by achieving relatively high accuracy on both the original and OOD datasets. They also exhibit minimal OOD performance degradation, signifying strong generalization capabilities and robustness to distributional changes.

**Scenario vs. Attack.** Table 3 demonstrates that Scenario-level and Attack-level perturbations generally degrade model performance, with Scenario-level perturbations often causing larger gaps due to the fine-grained semantic changes (altering scenarios while preserving core reasoning). For

Table 3: Performance on Scenario-level semi-fact data and Attack-level semi-fact data. Each performance value represents the accuracy in solving problems from the dataset. The accuracy of performance on attack-level semi-factual data is determined by computing the minimum accuracy across three different attacks.

| | AIME 2024 | | | AIME-500 | | |
|---|---|---|---|---|---|---|
| | Original | OOD (Scenario) | OOD (Attack) | Original | OOD (Scenario) | OOD (Attack) |
| o1-preview | 0.500 | 0.500 | 0.467 | 0.742 | 0.638 | 0.680 |
| o1-mini | 0.567 | 0.600 | 0.600 | 0.864 | 0.756 | 0.750 |
| o3-mini | 0.767 | 0.667 | 0.767 | 0.922 | 0.848 | 0.820 |
| Deepseek-R1 | 0.800 | 0.733 | 0.767 | 0.920 | 0.816 | 0.840 |
| GPT-4o | 0.133 | 0.100 | 0.100 | 0.278 | 0.204 | 0.190 |
| Deepseek-V3 | 0.367 | 0.333 | 0.333 | 0.528 | 0.438 | 0.420 |
| Mixtral-8x7B-IT-v0.1 | 0.000 | 0.000 | 0.000 | 0.012 | 0.000 | 0.012 |
| Qwen2.5-72B-IT | 0.200 | 0.167 | 0.200 | 0.432 | 0.290 | 0.392 |
| Qwen2.5-Math-72B-IT | 0.267 | 0.233 | 0.200 | 0.536 | 0.360 | 0.500 |
| LLAMA3.1-70B-IT | 0.200 | 0.167 | 0.167 | 0.424 | 0.244 | 0.392 |
| s1.1-32B | 0.533 | 0.500 | 0.478 | 0.752 | 0.654 | 0.494 |
| Gemma-2-27B-IT | 0.033 | 0.033 | 0.000 | 0.062 | 0.028 | 0.062 |
| Gemma-2-9B-IT | 0.000 | 0.000 | 0.000 | 0.032 | 0.016 | 0.026 |
| LLAMA3.1-8B-IT | 0.000 | 0.033 | 0.000 | 0.132 | 0.074 | 0.100 |
| Phi-3-mini-4k-IT | 0.000 | 0.000 | 0.000 | 0.046 | 0.024 | 0.028 |
| LLAMA3.2-3B-IT | 0.033 | 0.033 | 0.033 | 0.122 | 0.066 | 0.108 |

example, on AIME-500, scenario-level rephrasing reduces accuracy from 0.742 to 0.638 for o1-preview, while Attack-level perturbation cause a drop from 0.742 to 0.680. While Scenario-level and Attack-level perturbations measure distinct aspects of robustness, they are complementary. Combining the performance of Scenario and Attack-level perturbations data into a semi-factual generation metric provides a more robust framework for analyzing models' practical problem-solving capabilities.

### 4.2.2 Science Questions

Table 2 and Figure 5 show ID-OOD performance on GPQA Diamond dataset. The distribution of model performance points indicates that most models are more dispersed on this graph, suggesting that the difficulty of the GPQA Diamond dataset is less than that of AIME-500 and AIME 2024.

Models such as Mixtral-8x7B-Instruct-v0.1 and LLAMA3.2-3B-IT demonstrate relatively low accuracy levels. Notably, LLAMA3.2-3B-IT shows a 4.9% improvement in OOD performance compared to its ID performance, as shown in Table 2. This phenomenon can be attributed to the model's inherently poor performance. For the difficult questions in the GPQA Diamond dataset, the model seems to resort to random guessing, resulting in an accuracy that fails to reach the 25% level expected from random guessing. Consequently, the OOD performance reflecting a higher accuracy than ID is a result of the same guessing strategy employed for both ID and OOD questions.

## 5 Analysis and Discussion

This section presents a detailed analysis of OOD robustness of reasoning and non-reasoning models, addressing open research questions that are fundamental to building efficient reasoning models.

**RQ1: Do reasoning models deliver the significant performance and robustness improvements claimed in their reports?**

Overall, the o1-series models, along with o3, Deepseek-R1, and s1, exhibit strong performance in complex logical reasoning tasks, such as AIME and GPQA Diamond, on the original datasets. While performance declines from ID to OOD across various datasets, these models maintain notable robustness. Their improved performance, consistent with the claims in their reports OpenAI [2024, 2025], Guo et al. [2025], Muennighoff et al. [2025], suggests that these advancements are reliable.

Notably, on AIME-500 and AIME 2024, while o3-mini and Deepseek-R1 exhibit a larger ID-OOD performance gap than o1-preview, they still achieve the highest absolute performance in the OOD setting. This superior mathematical reasoning capability is likely due to their exposure to a larger corpus of mathematical problems during training, enhancing their problem-solving abilities. On

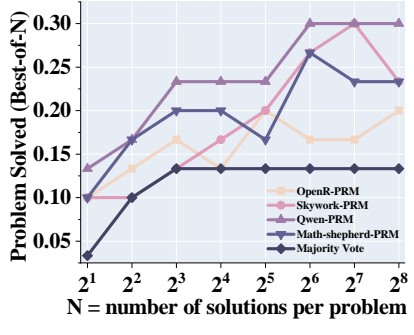
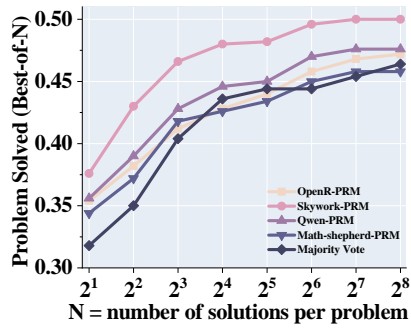

(a) Performance on AIME 2024 OOD data.  (b) Performance on AIME-500 OOD data.

Figure 4: **Test-time Scaling Law.** We show that the model's performance increases on the OOD dataset with the test-time computation budget increases using Qwen2.5-Math-7B-IT as the policy model, along with several PRMs.

Table 4: Performance comparison of different process reward models using Qwen2.5-Math-7B-IT under the Best-of-256 test. All methods utilize Qwen2.5-Math-7B-IT as the base model. Qwen2.5-Math-7B-IT's performance represents majority vote (Maj@256) results, while other methods are based on PRMs.

|  | **AIME 2024** | | **AIME-500** | |
|---|---|---|---|---|
|  | Original | OOD | Original | OOD |
| Qwen2.5-Math-7B-IT (Maj@256) | 0.167 | 0.133 | 0.524 | 0.464 |
| +Math-Shepherd | 0.233 | 0.233 | 0.528 | 0.458 |
| +OpenR | 0.233 | 0.200 | 0.526 | 0.472 |
| +Skywork | 0.200 | 0.233 | 0.582 | 0.500 |
| +Qwen | 0.300 | 0.300 | 0.538 | 0.476 |

GPQA Diamond, o3-mini and Deepseek-R1 also achieve the highest performance, underscoring their general reasoning advantages.

**RQ2: Is there a possibility of data leakage in the original datasets for AIME-500?**

Analyzing the ID-OOD performance gap, as depicted in Figure 1, reveals a notable difference between AIME 2024 and AIME-500. While the gap between ID and OOD performance is relatively small for AIME 2024, AIME-500 consistently shows a larger gap. This observation suggests that a significant portion of the models may have encountered similar AIME-500 data during training. Given that AIME data prior to 2024 was publicly available, we posit that data leakage is likely for most models.

**RQ3: Can our OOD dataset serve as a test-time computation benchmark?**

Based on our analysis, the AIME-500, AIME 2024, and GPQA Diamond datasets present significant challenges and possess strong discriminative power. As shown in Figure 4, model performance improves with increased test-time computation, underscoring the quality of our data. The upward trend and variation among the five lines indicate the dataset's ability to differentiate model performance during test-time computation.

Furthermore, Table 4 presents results from 256 inferences using Qwen2.5-Math-7B-IT to evaluate different PRMs. For AIME 2024, OOD performance shows slight variations compared to ID across PRMs. However, for AIME-500, all PRMs experience a notable decline in OOD performance, suggesting possible exposure to similar problems before 2024, and thus indicating data leakage. In contrast, AIME 2024 data appears largely unseen by the models in terms of the low-level performance decay. Our methodology provides valuable insights into assessing robust LLM reasoning. The dynamic generation capability allows model validation, preventing exploitation or overfitting, which can serve as an effective benchmark for facilitating future research on test-time computation.

## 6 Limitations

While ThinkBench presents a systematic dynamic OOD data generation framework that uniquely decouples reasoning robustness from memorization biases, it still has some limitations. First, due to cost constraints, Scenario-level semi-fact generation adopts a single-path rephrasing strategy instead of hierarchical multi-scenario branching, limiting OOD diversity. Future work needs to consider expanding task coverage and integrating tree-structured generation for richer scenario variations. It is worth emphasizing that ThinkBench's dynamic generation method possesses inherent scalability: by repeatedly applying our generation pipeline to the same original problem, a large number of OOD variants with different scenarios and expressions can be created. Second, while ThinkBench is designed to advance robust evaluation of LLM reasoning capabilities, we acknowledge that potential misuse risks may be associated with its dynamic data generation methodology. We emphasize that responsible use of ThinkBench is crucial for ensuring its value as a research tool while minimizing potential security and privacy risks.

## 7 Ethics Statement

The development of ThinkBench adheres to principles of responsible research and ethical AI practices.

**Data Privacy and Security.** All datasets used in experiments are publicly available and licensed for research. We verified that our OOD data generation process does not introduce personally identifiable information or sensitive content. For future deployments, we recommend that implementers integrate privacy-preserving mechanisms, particularly when handling proprietary or sensitive data.

**Responsible Release and Usage.** To prevent potential misuse, ThinkBench is released under a restrictive license that explicitly prohibits the following uses: (1) reverse-engineering LLM training data; (2) conducting membership inference or privacy attacks; (3) maliciously probing deployed systems to discover vulnerabilities; (4) deployment in high-stakes applications without appropriate safety measures. Users need to agree to these terms before using ThinkBench.

ThinkBench contributes positively to the development of trustworthy AI systems by providing a reliable method for evaluating LLM reasoning capabilities and identifying data contamination issues.

## 8 Conclusion

We presented ThinkBench, a first robust dynamic evaluation benchmark for testing reasoning capability in LLMs, unifying the reasoning models and non-reasoning models evaluation. ThinkBench offers dynamic data generation to evaluate the out-of-distribution of models through various math reasoning, and science questions samples. Experiments over AIME-500, AIME 2024, and GPQA Diamond indicate that while reasoning models represented by o1 and o3 consistently demonstrate relatively strong robustness, most LLMs fall short of robust performance. Experiments on ThinkBench highlight the importance of eliminating data contamination, especially in reasoning tasks where golden answers are prone to leakage. ThinkBench offers an effective solution to mitigate data leakage issues during reasoning evaluation. For future work, ThinkBench's dynamic data construction can be scaled to datasets of arbitrary size for systematic data contamination detection. Furthermore, by iteratively applying the construction pipeline to the same problem, researchers can generate diverse rephrasings across varied scenarios and expressions, expanding the scale and coverage of reasoning evaluation.

## Acknowledgments and Disclosure of Funding

This work has been financially supported by the National Key R&D program of China No. 2022YFE0204900 and the National Natural Science Foundation of China (NSFC) Key Project under Grant Number 62336006.

We thank all participants who contributed to ThinkBench.

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

Table 5: Comparison with OOD Benchmarks. Reasoning tasks include Math, Physics, Chemistry, and Biology problems. Eval represents evaluation.

| | GLUE-X | PromptBench | CofQA | ThinkBench |
|---|---|---|---|---|
| **Perturbation Granularity** | - (use other samples) | Attack-level | Entity-level | Scenario-level and Attack-level |
| **Factual Perturbations** | - | - | Counterfact | Semi-fact |
| **Non-Reasoning Models Eval** | ✓ | ✓ | ✓ | ✓ |
| **Reasoning Models Eval** | × | × | × | ✓ |
| **Tasks** | NLU | NLU and Translation | QA | Reasoning Tasks |

# A Dataset

## A.1 Comparative Analysis with Existing OOD Benchmarks

Unlike prior works that focus on syntactic or adversarial noise, ThinkBench is the first to systematically alter problem scenarios, introducing Scenario-level Semi-fact Data, which rephrases problems into entirely new contexts while preserving core reasoning logic. This tests generalization beyond memorization. Our semi-factual data integrates innovative scenario-level and attack-level perturbations, which provide a more robust approach to evaluating reasoning models. We also provide a robust evaluation for both Reasoning Models and Non-Reasoning Models. The differences between ThinkBench and other typical OOD benchmarks are shown in Table 5.

## A.2 Case Study

Table 12, Table 13 and Table 14 show the case of different models tackling ThinkBench, including question, choices, answer in Original set and OOD set, and responses from models.

Through the example in Table 12, we observe that our dynamic construction transforms the original question into a new question, with options also being rephrased and reordered. Our benchmark reduces the likelihood of models achieving high scores through rote memorization. In responses from GPT-4o and Llama3.1-70B-IT, it is evident that while both models correctly answer the original question, they fail to provide correct responses to the corresponding OOD data. Both models provide detailed analyses and correct answers to the original test data. However, for the OOD data, Llama3.1-70B-IT does not offer a detailed analysis and instead directly gives an incorrect answer. Although GPT-4o conducts some analysis, it confuses the concepts of "Kantianism" and "contractualism", even mixing them up during the reasoning process, ultimately leading to an incorrect answer.

## A.3 Failure Case Analysis

We present a thorough investigation of failure cases encountered during our previous attempt.

For instance, the original problem was: "There exist real numbers $x$ and $y$, both greater than 1, such that $\log_x(y^x) = \log_y(x^{4y}) = 10$. Find $xy$." After rephrasing and modifying the scenario, it was incorrectly rewritten as: "In a magical garden, there are two unique types of flowers whose quantities are given by real numbers $x$ and $y$, where $a > 1$ and $b > 1$. They follow a special logarithmic relationship $\log_a(b^a) = \log_b(a^{4b}) = 10$. Find $xy$." This type of failure case arises from contextual variable confusion, which causes the evaluation model to be uncertain about what to compute, leading to incorrect answers. As a result, it fails to fairly assess the model's reasoning capability.

Notably, ThinkBench is able to effectively filter such failure cases through iterative verifier's validation, ensuring contextual coherence and consistency. The high valid scores (1.00) across all evaluated datasets in Section A.5 demonstrate that our framework successfully maintains high quality and logical validity while introducing expression and scenario distributional shifts.

## A.4 Semi-fact data Construction Cost

The OOD generation pipeline uses GPT-4o for both rephrasing and verification. Each scenario-level instance requires 3x GPT-4o calls due to iterative regeneration if verification fails, while attack-level perturbations require 1.25x due to simpler perturbations.

Table 6: Scores of Human Evaluation across datasets.

| Datasets | Logical Equivalence Score | Question Similarity Score | Valid Score | Fleiss' kappa |
|---|---|---|---|---|
| AIME 2024 | 0.84 | 0.36 | 1.00 | 0.83 |
| AIME-500 | 0.88 | 0.38 | 1.00 | 0.77 |
| GPQA Diamond | 0.80 | 0.49 | 1.00 | 0.81 |
| MMLU-570 | 0.94 | 0.54 | 1.00 | 0.90 |

Table 7: Scores of Automated Evaluation across datasets.

| Datasets | Logical Equivalence Score | Question Similarity Score | Valid Score |
|---|---|---|---|
| AIME 2024 | 0.92 | 0.43 | 1.00 |
| AIME-500 | 0.87 | 0.48 | 1.00 |
| GPQA Diamond | 0.95 | 0.55 | 1.00 |
| MMLU-570 | 0.97 | 0.57 | 1.00 |

## A.5 Validation of Datasets' High Quality

We recruit six annotators (three undergraduates and three graduates) for further validation of our constructed data. To ensure evaluation fairness, annotators are not informed which instances are original versus OOD data. They are instructed to assess: (1) consistency between corresponding questions, (2) validity of question-answer pairs (i.e., the answer correctly addressed the question), and (3) logical equivalence scoring (0: completely non-equivalent, 0.5: partially equivalent, 1: fully equivalent) and similarity scoring (0: completely dissimilar, 0.5: partially similar, 1: nearly identical).

Beyond AIME 2024, we conduct a comprehensive human evaluation on AIME-500, GPQA Diamond, and MMLU-570, with 30 samples annotated per dataset. We also report Fleiss' Kappa values to measure annotation reliability.

We complement human evaluation with DeepSeek-R1 automated scoring using similar prompts, achieving consistent results that validate our human annotations.

As shown in Table 6 and Table 7, perfect Valid Scores (1.00) across all datasets confirm that constructed questions and answers remain properly aligned. High logical equivalence scores demonstrate that our OOD data preserves the core logical structure while introducing irrelevant distractors.

It is also worth noting that higher question similarity is not necessarily better, as the intention of constructing semi-fact OOD data was to preserve the core knowledge of the questions while modifying their expression and scenarios.

## B ThinkBench on Language Understanding

As illustrated in Figure 6, we also present the results of dynamically constructing 570 MMLU OOD data using ThinkBench. There is a significant performance gap between Out-Of-Distribution (OOD) and In-Distribution (ID) data across different models. The o1-preview model demonstrates superior performance on this knowledge-intensive dataset. Furthermore, reasoning models like o1-preview and o1-mini exhibit greater robustness compared to non-reasoning models such as GPT-4o.

## C ThinkBench on Other Reasoning Tasks

In addition to the evaluation of AIME 2024, AIME-500, GPQA Diamond, and MMLU, we also implement our evaluation to include diverse reasoning types:

We sample 30 instances each from LogiQA (logical reasoning), ARC_Challenge (commonsense reasoning), and 270 instances from BBH (covering 27 reasoning subtasks, including symbolic reasoning). Using our dynamic data construction, we generate corresponding Semi-fact OOD data.

As shown in Table 8, models show varying performance degradation patterns: most pronounced on LogiQA (logical reasoning), minimal on ARC_Challenge (commonsense reasoning), and moderate on BBH (multiple reasoning tasks). This suggests current models possess reasonable commonsense

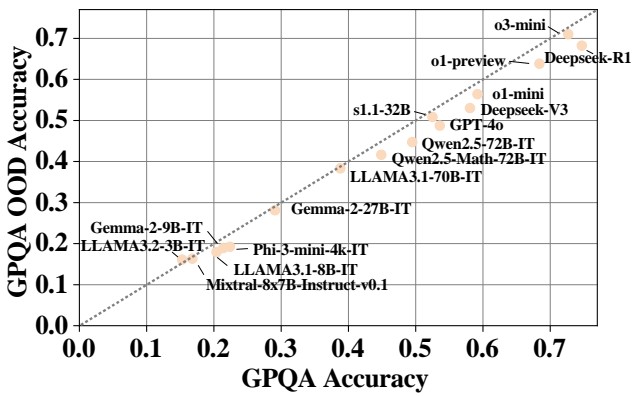

Figure 5: Scientific Questions Gap on GPQA Diamond.

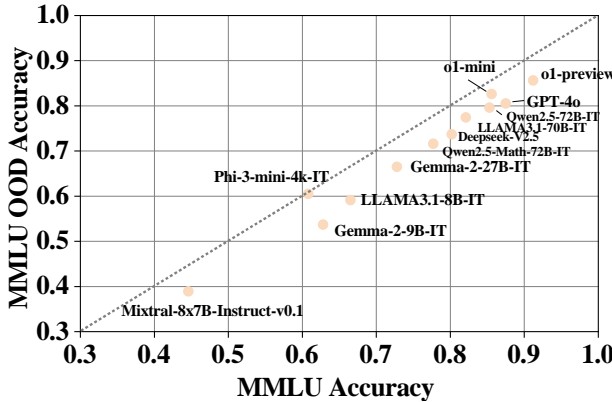

Figure 6: Language Understanding Gap on MMLU 570.

capabilities but struggle with complex logical reasoning, often relying on memorized patterns rather than genuine understanding.

Qwen2.5-Math-72B-IT demonstrates superior performance and robustness on logical and symbolic reasoning tasks compared to the general Qwen2.5-72B-IT, while showing comparable performance on commonsense reasoning. This indicates that mathematical training enhances logical reasoning capabilities but provides limited benefits for commonsense reasoning.

These comprehensive evaluations across diverse reasoning types demonstrate the robustness and generalizability of our dynamic OOD construction.

Table 8: Model performance comparison across different reasoning datasets. We report Ori(ginal) and OOD performance on each reasoning dataset.

| Models | LogiQA (Ori) | LogiQA (OOD) | ARC_C (Ori) | ARC_C (OOD) | BBH (Ori) | BBH (OOD) |
|---|---|---|---|---|---|---|
| Deepseek-R1 | 0.833 | 0.867 | 1.000 | 0.984 | 0.856 | 0.815 |
| s1.1-32B | 0.800 | 0.517 | 0.967 | 0.967 | 0.737 | 0.741 |
| Deepseek-V3 | 0.767 | 0.633 | 0.967 | 0.933 | 0.811 | 0.767 |
| Qwen2.5-Math-72B-IT | 0.700 | 0.567 | 0.900 | 0.833 | 0.804 | 0.811 |
| Qwen2.5-72B-IT | 0.700 | 0.517 | 0.933 | 0.900 | 0.752 | 0.700 |
| LLAMA3.1-70B-IT | 0.633 | 0.667 | 0.867 | 0.750 | 0.678 | 0.652 |
| LLAMA3.1-8B-IT | 0.367 | 0.367 | 0.500 | 0.567 | 0.507 | 0.493 |

Table 9: Error distributions on Original and Scenario-level OOD data.

| Error Type | Original | | Scenario-level OOD | |
|---|---|---|---|---|
| | o3-mini | LLAMA3.1-70B-IT | o3-mini | LLAMA3.1-70B-IT |
| Lack of Analysis | 0 | 19 | 0 | 42 |
| Hallucination | 17 | 66 | 13 | 83 |
| COT breakdown | 8 | 87 | 22 | 104 |
| Numerical Error | 7 | 38 | 12 | 53 |
| Wrong Approach Selection | 4 | 33 | 8 | 57 |
| Other | 3 | 45 | 21 | 39 |
| Total | 39 | 288 | 76 | 378 |

## D Fine-grained Analysis of Data Leakage

To provide more granular insights into data leakage, we conduct a comprehensive error analysis. We categorize errors into six types: Lack of Analysis (direct answers without reasoning), Hallucination (introducing non-existent conditions), Wrong Approach Selection (incorrect methodological choices), CoT Breakdown (reasoning chain failures), Numerical Error (computational mistakes), and Other.

Using DeepSeek-R1 annotation, we analyze error distributions on AIME-500. As illustrated in Table 9, LLAMA3.1-70B-IT shows a significant increase in "Lack of Analysis" errors on OOD data, compared to the original data (19 vs. 42), indicating the model becomes unfamiliar with scenario-shifted problems and skips detailed reasoning. This suggests a substantial data leakage issue in the original AIME-500 problem scenarios for LLAMA3.1-70B-IT. Both models exhibit notable increases in "Wrong Approach Selection" and "COT Breakdown" errors on OOD data, compared to the original data. It suggests that memorized approaches and CoT solutions from familiar questions cannot be directly applied to unfamiliar scenarios. These patterns expose substantial data leakage in both original question's scenarios and expressions, confirming that models rely heavily on memorized solutions rather than genuine reasoning capabilities.

## E Comparison with Naturally Occurring OOD Data

To further explore the quality of our constructed data, we use the newly published real exam AIME 2025 and compare model performance on AIME 2025 (natural OOD) against our constructed AIME 2024 OOD data.

As shown in Table 10, naturally occurring OOD (AIME 2025) causes more severe performance drops than our constructed OOD (e.g., for DeepSeek-R1: from 0.800 to 0.667 vs. from 0.800 to 0.750). While the degradation from our data is less severe, it is still close to that of natural OOD. This suggests our automatically generated data, while not as high-quality as human-created OOD, is still effective for identifying and mitigating data leakage.

Table 10: Performance comparison of constructed and naturally occurring OOD data.

| Models | AIME 2024 | AIME 2024 (OOD) | AIME 2025 |
|---|---|---|---|
| Deepseek-R1 | 0.800 | 0.750 | 0.667 |
| s1.1-32B | 0.533 | 0.517 | 0.500 |
| Deepseek-V3 | 0.367 | 0.333 | 0.333 |
| Qwen2.5-Math-72B-IT | 0.267 | 0.217 | 0.233 |
| Qwen2.5-72B-IT | 0.200 | 0.184 | 0.133 |
| LLAMA3.1-70B-IT | 0.200 | 0.167 | 0.133 |
| LLAMA3.1-8B-IT | 0.000 | 0.017 | 0.033 |

# F   Performance of Adaptive Method

We also explore the performance of adaptive baseline on AIME-500 and AIME 2024. Specifically, we use prompt-based planning (+Plan) to enhance OOD performance.

Table 11: The performance of the adaptive method. We report Ori(ginal) and OOD performance.

| Models | AIME 2024 (Ori) | AIME 2024 (OOD) | AIME-500 (Ori) | AIME-500 (OOD) |
|---|---|---|---|---|
| Qwen2.5-Math-72B-IT | 0.267 | 0.217 | 0.536 | 0.430 |
| Qwen2.5-Math-72B-IT+Plan | 0.233 | 0.234 | 0.552 | 0.463 |
| Qwen2.5-72B-IT | 0.200 | 0.184 | 0.432 | 0.341 |
| Qwen2.5-72B-IT+Plan | 0.200 | 0.167 | 0.438 | 0.377 |
| LLAMA3.1-70B-IT | 0.200 | 0.167 | 0.424 | 0.318 |
| LLAMA3.1-70B-IT+Plan | 0.233 | 0.184 | 0.426 | 0.355 |
| LLAMA3.1-8B-IT | 0.000 | 0.017 | 0.132 | 0.087 |
| LLAMA3.1-8B-IT+Plan | 0.033 | 0.017 | 0.108 | 0.100 |

As shown in Table 11, on both datasets, the adaptive method (+Plan) does not significantly improve original data performance. It generally exhibits a smaller performance drop on OOD data compared to non-adaptive methods. This shows the adaptive approach can partially mitigate OOD degradation. However, a noticeable performance drop still remains on OOD compared to the original, proving our dynamically constructed OOD data is still a challenging and effective, even for the adaptive method. This confirms our data's high quality in exposing data leakage.

# G   Process Reward Models

In the past, language model training primarily used Outcome-based Reinforcement Models (ORM) [Wang et al., 2024a]. A foundational example is the ORM-based model by Cobbe et al. [2021], which focuses on training evaluators to assess the correctness of answers, providing crucial feedback. Meanwhile, the Process Reward Model (PRM) aims to provide stepwise rewards, offering fine-grained supervision. DeepMind [Uesato et al., 2022] supervises both reasoning steps and final results, while OpenAI [Lightman et al., 2023] introduces PRM800K, a human-annotated dataset, emphasizing step verification. Li et al. [2022] enhance result reliability with evaluator models and majority voting. Yu et al. [2024] improve reasoning through reinforcement learning with outcome and process supervision. The Generative Reward Model (GenRM)[Zhang et al., 2024] allows rich interaction between evaluators and generators, reflecting a demand for sophisticated process supervision. Recent work [Zheng et al., 2024] offers a benchmark for evaluating error identification in mathematical reasoning, fostering scalable oversight research. As a supplement, we propose a dynamic benchmark for testing reasoning capability in LLMs with the help of PRMs.

## G.1   Process Reward Models on Test-time Computation Budget

### G.1.1   Settings

To comprehensively conduct the test-time scaling evaluation for the various PRMs on AIME 2024 and AIME-500, we employ these PRMs:

(1) **Math-Shepherd-PRM**: The scoring mechanism employed in Math-Shepherd is essential for understanding the performance of reasoning steps. In this framework, the token "kn" is used to indicate the position where the step score is predicted. A "+" token represents a good step, one that contributes positively towards reaching the correct answer, while a "-" token signals a bad step. Notably, during the training of PRMs, the loss is computed only at positions marked with "kn". The PRM denoted as $(P \times S \to \mathbb{R}^+)$, assigns a score to each reasoning step $s$. The model is typically trained using the following binary cross-entropy loss function:

$$\mathcal{L}_{PRM} = \sum_{i=1}^{K} y_{s_i} \log r_{s_i} + (1 - y_{s_i}) \log(1 - r_{s_i}), \tag{2}$$

where $y_{s_i}$ represents the ground-truth label of the $i$-th reasoning step $s_i$, $r_{s_i}$ is the sigmoid output score predicted by the PRM for step $s_i$, and $K$ is the total number of reasoning steps in a given

solution $s$. To estimate the quality of each reasoning step, two methods are employed: hard estimation (HE) and soft estimation (SE). In HE, a step is considered good if it contributes to reaching the correct answer $a^*$, as defined by:

$$y_{s_i}^{HE} = \begin{cases} 1, & \text{if } \exists a_j \in A, a_j = a^* \\ 0, & \text{otherwise.} \end{cases} \tag{3}$$

In contrast, SE estimates the quality of a step based on the frequency with which it leads to the correct answer across multiple attempts:

$$y_{s_i}^{SE} = \frac{\sum_{j=1}^{N} \mathbb{I}(a_j = a^*)}{N}, \tag{4}$$

where $\mathbb{I}(a_j = a^*)$ is an indicator function that returns 1 if the $j$-th attempt $a_j$ equals the correct answer $a^*$, and $N$ is the total number of attempts. Once the labels for each reasoning step are obtained, the PRM is trained using the cross-entropy loss function to optimize the model's ability to correctly classify each step as either good or bad (Wang et al. [2024c]).

(2) **OpenR-PRM**: The Problem Resolution Model (PRM) computes a score, denoted as $p_t$, based on the current problem $q$ and the sequence of solution steps up to time $t$, represented as $[x_1, \ldots, x_t]$. This approach allows for a precise and detailed analysis of the solution process, helping identify errors as they occur, as noted by Lightman et al. [2023]. The main goal of PRMs is to assess whether the solution process is on track, with a score $y_t$ calculated to represent the correctness of the solution at step $t$, ranging from 0 to 1. This score is given by $y_t = \text{PRM}(q, x_1, x_2, \ldots, x_t)$. During training, the model is framed as a next-token prediction task, where labels are assigned as correct or incorrect, represented by positive (+) or negative (-) tokens. After training, OpenR uses the PRM to evaluate the correctness of each solution step during inference, assigning a score $r_{PRM_t}$ to each step. Two strategies are employed to compute a final score: the PRM-Min strategy, which selects the minimum score among all steps, $v = \min\{r_{PRM_t}\}_{t=0}^{T}$, and the PRM-Last strategy, which takes the score of the final step, $v = r_{PRM_T}$. PRMs function as dense verifiers, providing strong feedback that can significantly enhance the overall solution process (Wang et al. [2024a]).

(3) **Skywork-PRM**: We evaluate Skywork o1 Open-PRM-Qwen-2.5-7B o1 Team [2024], which extends the capabilities of the 1.5B model by scaling up to handle more demanding reasoning tasks, pushing the boundaries of AI reasoning.

(4) **Qwen-PRM**: We evaluate 7B version of Qwen-PRM Zhang et al. [2025]. This is a process reward model designed to offer feedback on the quality of reasoning and intermediate steps in mathematical problems. It is part of the Qwen2.5-Math series and has shown impressive performance in identifying errors in reasoning processes.

As shown in Figure 4, the model's performance improves on the AIME 2024 OOD and AIME-500 OOD dataset as the test-time computation budget increases and the best-of-N performance of each reward model varies as a function of N. Since majority voting is known to be a strong baseline(Lewkowycz et al. [2022], Wang et al. [2022]), we also include this method as a point of comparison. We compared a series of PRM methods and then observed the performance changes of various PRM methods as N increased.

Skywork-PRM and Qwen-PRM achieve relatively high performance across all computation budgets. OpenR shows moderate performance, improving steadily with increasing N, but its improvement rate is slower compared to Skywork-PRM and Qwen-PRM. Majority Voting mainly underperforms other methods, even with increased computation budgets, showing limited scalability. The possible underlying reasons are these: (1)Model-specific Capabilities: The superior performance of Skywork-PRM and Qwen-PRM may result from their advanced mechanisms for aggregating or utilizing the increased number of solutions, likely benefiting from better exploration of the solution space or more robust voting schemes. (2)Algorithmic Limitations: The suboptimal performance of Majority Voting suggests its inability to effectively utilize additional solutions. (3)Dataset Complexity: The AIME 2024 OOD dataset and AIME-500 OOD dataset likely require nuanced reasoning and adaptability, favoring methods like Skywork-PRM and Qwen-PRM that can better handle out-of-distribution (OOD) generalization tasks. Thus, the results highlight the importance of adopting advanced techniques that can efficiently utilize increased test-time computation budgets. Methods like Skywork-PRM and

Table 12: Case Study on MMLU 570.

**Original Question**: Carruthers approaches questions about the moral standing of nonhuman animals from the perspective of which of the following moral theories?
**Original Choices**: A: consequentialism B: Kantianism C: virtue ethics D: contractualism
**Original Answer**: D
**OOD Question (Scenario-level Semi-fact Question)**: In his exploration of ethical frameworks, Carruthers considers various moral theories before honing in on a particular approach to evaluate the moral status of nonhuman animals. From which moral theory does Carruthers analyze the ethical significance of these animals?
**OOD Choices**: A: ethics of character B: contractualism C: outcomes-based ethics D: Kantianism
**OOD Answer**: B

| **GPT-4o on Original:** | **Llama3.1-70B-IT on Original:** |
|---|---|
| Peter Carruthers is known for approaching questions about the moral standing of nonhuman animals from the perspective of contractualism. Contractualism is a moral theory that determines moral principles based on what rational agents would agree upon as part of a social contract. Carruthers argues that nonhuman animals do not have direct moral standing within contractualism because they are not capable of participating in the contract-making process. Therefore, the correct answer is **D**. | Carruthers approaches questions about the moral standing of nonhuman animals from the perspective of contractualism. Contractualism is a moral theory that emphasizes the importance of mutual agreements and contracts in determining moral obligations. In the context of nonhuman animals, contractualism would suggest that animals have moral standing because they are capable of participating in a "contract" or social agreement with humans, even if it is not a formal or explicit one. The correct answer is **D**. |
| **GPT-4o on OOD:** | |
| Carruthers is known for utilizing Kantianism as a framework for evaluating the moral status of nonhuman animals. This approach emphasizes the importance of rationality and moral agency, aligning with Kantian ethical principles. Therefore, the correct answer is **D**. | **Llama3.1-70B-IT on OOD:** Based on the information provided, the correct answer is **C**: outcomes-based ethics. |

Qwen-PRM demonstrate superior scaling behavior, suggesting their robustness and adaptability in solving complex problems.

**Original Question**: There exist real numbers $x$ and $y$, both greater than 1, such that $\log_x(y^x) = \log_y(x^{4y}) = 10$. Find $xy$.

**Original Answer**: 25

**OOD Question (Scenario-level Semi-fact Question)**: Consider two species of plants with growth rates represented by real numbers $x$ and $y$, both exceeding 1, such that the growth rate of one species raised to the power of the other results in the equation $\log_x(y^x) = \log_y(x^{4y}) = 10$. Determine the product of their growth rates, $xy$.

**OOD Answer**: 25

**GPT-4o on Original:**

$$\log_x(y^x) = 10$$
$$\log_y(x^{4y}) = 10$$

We'll solve for $xy$. Starting with the first equation:

$$\log_x(y^x) = 10$$

This can be rewritten using the logarithm power rule:

$$x \log_x y = 10$$

Thus,

$$\log_x y = \frac{10}{x}$$

The base change formula $\log_x y = \frac{\log y}{\log x}$ gives us:

$$\frac{\log y}{\log x} = \frac{10}{x}$$

So,

$$\log y = \frac{10}{x} \log x$$

Now, for the second equation:

$$\log_y(x^{4y}) = 10$$

This expands to:

$$4y \log_y x = 10$$

Thus,

$$\log_y x = \frac{10}{4y} = \frac{5}{2y}$$

The base change formula $\log_y x = \frac{\log x}{\log y}$ gives us:

$$\frac{\log x}{\log y} = \frac{5}{2y}$$

So,

$$\log x = \frac{5}{2y} \log y$$

Now substitute $\log y$ from the first equation into this:

$$\log x = \frac{5}{2y} \times \frac{10}{x} \log x$$

Simplifying,

$$\log x = \frac{50}{2xy} \log x$$

Cancel $\log x$ from both sides (assuming $\log x \neq 0$):

$$1 = \frac{50}{2xy}$$

Thus,

$$2xy = 50$$

So,

$$xy = 25$$

The value of $xy$ is **25**.

**GPT-4o on OOD:**

Step 1: Solve the first equation
The first equation is:

$$\log_x(y^x) = 10$$

This implies:

$$y^x = x^{10}$$

Step 2: Solve the second equation
The second equation is:

$$\log_y(x^{4y}) = 10$$

This implies:

$$x^{4y} = y^{10}$$

Step 3: Express $y$ in terms of $x$ from the first equation
From $y^x = x^{10}$, we can express $y$ as:

$$y = x^{10/x}$$

Step 4: Substitute $y$ in the second equation
Substitute $y = x^{10/x}$ into the second equation $x^{4y} = y^{10}$:

$$x^{4(x^{10/x})} = (x^{10/x})^{10}$$

Simplifying the right side:

$$x^{4(x^{10/x})} = x^{100/x}$$

Step 5: Equate the exponents
Since the bases are the same, equate the exponents:

$$4 \cdot x^{10/x} = \frac{100}{x}$$

Step 6: Solve for $x$
Rearrange and simplify:

$$4x^{10/x} \cdot x = 100$$
$$4x^{1+10/x} = 100$$
$$x^{1+10/x} = 25$$

Step 7: Solve for $x$ using trial and error or numerical methods
Let $z = 1 + \frac{10}{x}$, so $x^z = 25$. We need to find $x$ such that:

$$z = 1 + \frac{10}{x} \quad \text{and} \quad x^z = 25$$

By trial and error or numerical methods, you can find that $x = 5$ satisfies the equation.

Step 8: Find $y$
Now, substitute $x = 5$ back into $y = x^{10/x}$:

$$y = 5^{10/5} = 5^2 = 25$$

Step 9: Calculate the product $xy$
Finally, calculate the product:

$$xy = 5 \cdot 25 = 125$$

Thus, the product of their growth rates is **400**.

**Original Question**: cyclohexanone is treated with LDA at low temperature, followed by benzaldehyde. the reaction mixture is then acidified, forming product 1. 1 is then treated with an excess of diethylaminosulfur trifluoride, forming product 2. what is the structure of prodcut 2?

**Original Choices**: A: ((R)-((R)-2,2-difluorocyclohexyl)fluoromethyl)benzene B: (S)-2-((R)-fluoro(phenyl)methyl)cyclohexan-1-one C: (2R)-1-fluoro-2-((S)-fluoro(phenyl)methyl)cyclohexan-1-ol D: ((S)-((R)-2,2-difluorocyclohexyl)fluoromethyl)benzene

**Original Answer**: A

**OOD Question (Scenario-level Semi-fact Question)**: In the realm of organic synthesis, understanding the behavior of reagents under various conditions is crucial for designing effective synthetic routes. Let's consider a scenario where cyclohexanone, a common ketone, is reacted with lithium diisopropylamide (LDA), a strong base, at reduced temperature to form an enolate, which is then treated with benzaldehyde. This reaction typically forms an aldol product, which, upon acidification, yields a compound known as product 1. Further exploring the reactivity of such aldol products, if this product 1 is treated with an excess of diethylaminosulfur trifluoride, a reagent known for its ability to affect sulfur-based modifications in organic molecules, what would be the structure of the resulting product 2?

**OOD Choices**: A: ((S)-((R)-2,2-difluorocyclohexyl)fluoromethyl)benzene B: ((R)-((R)-2,2-difluorocyclohexyl)fluoromethyl)benzene C: (2R)-1-fluoro-2-((S)-fluoro(phenyl)methyl)cyclohexan-1-ol D: (S)-2-((R)-fluoro(phenyl)methyl)cyclohexan-1-one

**OOD Answer**: B

---

**o1-preview on Original:**

To solve this problem, let's break down the reactions step by step. **First Reaction:** Cyclohexanone is treated with LDA (Lithium Diisopropylamide) at low temperature. LDA is a strong, non-nucleophilic base that deprotonates the most accessible $\alpha$-hydrogen to form the kinetic enolate. In cyclohexanone, both $\alpha$-carbons are equivalent due to symmetry, so the enolate forms at one of these positions. The enolate then attacks benzaldehyde in an aldol addition reaction. The nucleophilic carbon of the enolate adds to the electrophilic carbonyl carbon of benzaldehyde, forming a new carbon-carbon bond. After protonation, this gives a $\beta$-hydroxy ketone (product 1). **Second Reaction:** Product 1, the $\beta$-hydroxy ketone, is treated with an excess of diethylaminosulfur trifluoride (DAST). DAST is a fluorinating agent that can convert alcohols to alkyl fluorides and ketones to geminal difluorides. In this case, DAST converts: - The hydroxyl group at the $\beta$-position into a fluorine atom. - The ketone group into a geminal difluoride. The overall result is a cyclohexane ring with: - Two fluorine atoms attached to C1 (from the ketone conversion). - A fluorine atom and a benzyl group attached to C2 (from the hydroxyl conversion). This gives us a product where: - C1 is now a difluoromethylene group (CF$_2$). - C2 has a fluorine and a benzyl group, with specific stereochemistry retained from the initial aldol addition. Considering the stereochemistry: - The initial aldol addition creates a new chiral center at C2. Since the starting materials are achiral and no chiral catalysts are used, a racemic mixture is formed. - The DAST reaction retains the configuration at C2 when converting the hydroxyl group to fluorine. - The former carbonyl carbon (C1) becomes a new chiral center after forming the geminal difluoride. From the options provided, option **A** corresponds to the product formed: - **Option A:** ((R)-((R)-2,2-difluorocyclohexyl)fluoromethyl)benzene This option indicates that: - The cyclohexane ring has two fluorines at C2 (now numbered as C1 in the name), making it a 2,2-difluorocyclohexyl group. - There is a fluoromethyl group attached to benzene, corresponding to the fluorine and benzyl group at C2. - Both chiral centers have the **R** configuration, matching the stereochemistry of the reactions. Thus, the correct answer is: **A**

---

**o1-preview on OOD:**

To solve this problem, let's systematically break down each step of the reaction and understand the transformations that occur. **Step 1: Formation of the Aldol Product (Product 1)** 1. **Formation of the Enolate Ion:** Cyclohexanone is treated with lithium diisopropylamide (LDA), a strong, non-nucleophilic base. LDA deprotonates the $\alpha$-carbon of cyclohexanone, forming the enolate ion. 2. **Aldol Addition Reaction:** The enolate ion is nucleophilic at the $\alpha$-carbon and attacks the electrophilic carbonyl carbon of benzaldehyde. This forms a new carbon-carbon bond between the $\alpha$-carbon of cyclohexanone and the carbonyl carbon of benzaldehyde. 3. **Acidification:** The alkoxide formed after the addition is protonated upon acidification, yielding a $\beta$-hydroxy ketone. The **structure of Product 1** is therefore a cyclohexanone ring with a hydroxyl group (OH) and a phenyl group (C$_6$H$_5$) attached to the same carbon (the $\alpha$-carbon). This gives us **2-hydroxy-2-phenylcyclohexanone**. **Step 2: Reaction with Diethylaminosulfur Trifluoride (DAST)** DAST is a fluorinating agent known to convert alcohols to alkyl fluorides (with inversion of configuration) and carbonyl groups to gem-difluorides. However, under standard conditions, ketones are less reactive than alcohols towards DAST, and selective fluorination of the alcohol can occur. In this case, when Product 1 is treated with excess DAST: 1. **Fluorination of the Alcohol:** The hydroxyl group is converted to a fluorine atom with inversion of configuration. This transforms the chiral center at the $\alpha$-carbon (which was R or S in the aldol product) into the opposite configuration due to the inversion during the SN2-like process. 2. **Ketone Retention:** The ketone group remains largely unreacted under these conditions because ketones are less reactive towards DAST compared to alcohols, especially in the presence of steric hindrance and electronic effects from adjacent groups (like the phenyl group). Thus, **Product 2** is **2-((R)-fluoro(phenyl)methyl)cyclohexan-1-one**, where the hydroxyl group has been replaced by a fluorine atom, and the configuration at the chiral center has inverted compared to Product 1. **Matching with the Given Options:** Option D corresponds exactly to this structure: - **(S)-2-((R)-fluoro(phenyl)methyl)cyclohexan-1-one** This indicates that the final product has: - A cyclohexanone ring (ketone at position 1) - A fluorine atom and a phenyl group attached to the same carbon at position 2 - Specific stereochemistry indicating the inversion at the $\alpha$-carbon Therefore, **Option D is the correct answer**.**D.**

