# OpenReview forum: "ThinkBench: Dynamic Out-of-Distribution Evaluation for Robust LLM Reasoning"
_NeurIPS.cc/2025/Datasets_and_Benchmarks_Track — NeurIPS 2025 Datasets and Benchmarks Track poster_

### Official Review · Reviewer_upCj · 2025-06-15

**Rating:** 5
**Confidence:** 2

**Summary:**

The paper presents ThinkBench, a benchmark for robust evaluation of reasoning in LLMs via dynamic, out-of-distribution (OOD) test sets. Using semi-factual scenario and attack perturbations, the framework decouples reasoning from memorization. Evaluations over math and science datasets confirm both data leakage in prior benchmarks and the superior robustness of models like o3 and Deepseek-R1.

**Additional Feedback:**

As I am not deeply familiar with evaluating papers in the benchmark track, I provide this review with low confidence. This paper indeed address the important problem of evaluting LLM robustness by out-of-distribution tests. And I have two additional questions.

Q1: In some cases, high performance on ThinkBench may be due to a model’s general robustness to input noise, rather than improved reasoning. Have the authors considered separating these dimensions in the evaluation?

Q2: OOD examples in ThinkBench are generated systematically. Is it possible to compare model performance on these with naturally occurring OOD errors (e.g., from newly published real exams or domain shifts)?

**Dataset Code Accessibility:**

Yes

**Dataset Code Comments:**

Code and pipeline are open-sourced. The protocol is modular and designed for community adoption.

**Ethical Considerations:**

No, there are no or only very minor ethics concerns

**Final Justification:**

The authors provide detailed experiments to support their findings, including various reasoning types, error types and baselines. This raise my confidence to keep my score at 5.

**Limitations Weaknesses:**

While AIME and GPQA tasks are valuable, the benchmark primarily focuses on STEM-style quantitative reasoning. Broader inclusion of symbolic logic, commonsense reasoning, or analogical reasoning would improve the generalizability of the benchmark.

The paper doesn't fully explain why certain models (e.g., LLM-as-PRM or open LLMs) fail under OOD variants. Some analysis of error types (e.g., hallucination, numerical error, chain-of-thought breakdown) would enhance interpretability and guide model improvement.

While model performance is thoroughly evaluated, the lack of adapted baselines (e.g., fine-tuned models on similar OOD tasks, or instruction-robust prompting strategies) makes it harder to contextualize the absolute performance drop.

**Strengths Contributions:**

The paper addresses a critical limitation in LLM evaluation: the lack of robust, dynamic out-of-distribution (OOD) benchmarks for reasoning tasks.

ThinkBench is built upon real-world math and science reasoning datasets (e.g., AIME 2024, GPQA Diamond), providing substantive tasks rather than synthetic toy examples. By targeting reasoning, rather than just classification robustness, the benchmark aligns well with current research priorities in LLM alignment and trustworthiness.

The paper reveals that many high-performing models on in-distribution (ID) datasets significantly underperform on OOD variants—demonstrating that ThinkBench is both non-trivial and diagnostic. Some smaller or more focused models (e.g., DeepSeek, o3 series) show better robustness, suggesting model specialization impacts reasoning generalization.

---

> ### Author Rebuttal · Authors · 2025-07-31
>
> We are sincerely grateful for your recognition and hope our reply solves your confusion.
>
> ## About Broader Inclusion of Reasoning Types
>
> We appreciate your suggestion to include more types of reasoning. We added evaluations on diverse reasoning tasks. We sampled 30 instances each from LogiQA (logical reasoning) and ARC_Challenge (commonsense reasoning), plus 270 instances from BBH (27 reasoning subtasks, including symbolic reasoning).
>
> Using our dynamic construction method, we generated corresponding Semi-fact OOD data, with the performance as follows:
>
> | Models              | LogiQA-30 (Original) | LogiQA-30 (OOD) | ARC_C-30 (Original) | ARC_C-30 (OOD) | BBH-270 (Original) | BBH-270 (OOD) |
> | ------------------- | -------------------- | --------------- | ------------------- | -------------- | ------------------ | ------------- |
> | deepseek-r1         | 0.833                | 0.867           | 1.000               | 0.984          | 0.856              | 0.815         |
> | s1.1-32B            | 0.800                | 0.517           | 0.967               | 0.967          | 0.737              | 0.741         |
> | deepseek-v3         | 0.767                | 0.633           | 0.967               | 0.933          | 0.811              | 0.767         |
> | Qwen2.5-Math-72B-IT | 0.700                | 0.567           | 0.900               | 0.833          | 0.804              | 0.811         |
> | Qwen2.5-72B-IT      | 0.700                | 0.517           | 0.933               | 0.900          | 0.752              | 0.700         |
> | LLAMA3.1-70B-IT     | 0.633                | 0.667           | 0.867               | 0.750          | 0.678              | 0.652         |
> | Llama-3.1-8B-IT     | 0.367                | 0.367           | 0.500               | 0.567          | 0.507              | 0.493         |
>
> Performance degradation varied: highest on LogiQA (logical), minimal on ARC_Challenge (commonsense). This suggests current models possess reasonable commonsense but struggle with complex logical reasoning, often relying on memorized patterns rather than genuine understanding.
>
> Compared to Qwen2.5-72B-IT, Qwen2.5-Math-72B-IT shows better performance and robustness on logical/symbolic tasks but comparable commonsense ability. This suggests more math training enhances logical reasoning but provides limited improvement in commonsense reasoning.
>
> These results confirm our dynamic OOD method is robust and generalizable. Our findings reveal data leakage in these tasks, and our method provides a fairer and more comprehensive evaluation framework.
>
> ## About Error Type Analysis
>
> We sincerely appreciate your concern about error types. The paper's Tables 6-8 show error cases from AIME 2024, GPQA Diamond, and MMLU. On original questions, Llama3.1-70B-IT provides detailed analysis and correct answers. On scenario-level OOD, however, it gives incorrect answers directly without analysis. Error cases in the paper also reveal chain-of-thought (CoT) breakdowns and numerical errors.
>
> To further analyze the error cases, we categorized errors into six types: Lack of Analysis (direct answers without reasoning), Hallucination (introduces non-existent conditions), Wrong Approach Selection (incorrect methodological choices), CoT Breakdown (reasoning chain failures), Numerical Error (computational mistakes), and Other.
>
> Using DeepSeek-R1 annotation, we show error distributions on AIME-500:
>
>  **Table1：Error distributions on the Original data**
>
> | **Models**      | **Lack of Analysis** | **Hallucination** | **COT breakdown** | **Numerical Error** | **Wrong Approach Selection** | **Other** | **Total** |
> | --------------- | -------------------- | ----------------- | ----------------- | ------------------- | ---------------------------- | --------- | --------- |
> | o3-mini         | 0                    | 17                | 8                 | 7                   | 4                            | 3         | 39        |
> | Llama3.1-70B-IT | 19                   | 66                | 87                | 38                  | 33                           | 45        | 288       |
>
>   **Table2：Error distributions on the Scenario-level OOD data**
>
> | **Models**      | **Lack of Analysis** | **Hallucination** | **COT breakdown** | **Numerical Error** | **Wrong Approach Selection** | **Other** | **Total** |
> | --------------- | -------------------- | ----------------- | ----------------- | ------------------- | ---------------------------- | --------- | --------- |
> | o3-mini         | 0                    | 13                | 22                | 12                  | 8                            | 21        | 76        |
> | Llama3.1-70B-IT | 42                   | 83                | 104               | 53                  | 57                           | 39        | 378       |
>
> - Llama3.1-70B-IT shows a significant increase in "Lack of Analysis" errors on OOD, compared to the original (19→42), indicating model becomes unfamiliar with scenario-shifted problems and skips detailed reasoning. This suggests a substantial data leakage issue in the original AIME-500 scenarios for Llama3.1-70B-IT.
> - Both models exhibit notable increases in "Wrong Approach Selection" and "COT Breakdown" errors on OOD, compared to the original. It suggests that memorized approaches and CoT solutions cannot be directly applied to unfamiliar scenarios.
>
> ## About Adaptive Baselines
>
> We appreciate your concern about the lack of adaptive baselines, and we added the adaptive baseline on AIME-500 and AIME 2024. Following [1], we used prompt-based planning (+Plan) to enhance OOD performance. Results for several models are below:
>
> | Models                   | AIME 2024 (Original) | AIME 2024 (OOD) | AIME-500 (Original) | AIME-500 (OOD) |
> | ------------------------ | -------------------- | --------------- | ------------------- | -------------- |
> | Qwen2.5-Math-72B-IT      | 0.267                | 0.217           | 0.536               | 0.430          |
> | Qwen2.5-Math-72B-IT+Plan | 0.233                | 0.234           | 0.552               | 0.463          |
> | Qwen2.5-72B-IT           | 0.200                | 0.184           | 0.432               | 0.341          |
> | Qwen2.5-72B-IT+Plan      | 0.200                | 0.167           | 0.438               | 0.377          |
> | Llama-3.1-70B-IT         | 0.200                | 0.167           | 0.424               | 0.318          |
> | Llama-3.1-70B-IT+Plan    | 0.233                | 0.184           | 0.426               | 0.355          |
> | Llama-3.1-8B-IT          | 0.000                | 0.017           | 0.132               | 0.087          |
> | Llama-3.1-8B-IT+Plan     | 0.033                | 0.017           | 0.108               | 0.100          |
>
> On both datasets, adaptive method (+Plan) does not significantly improve original data performance. It generally exhibits a smaller performance drop on OOD data compared to non-adaptive methods. This shows the adaptive approach can partially mitigate OOD degradation. However, a noticeable performance drop still remains on OOD compared to the original, proving our dynamically constructed OOD data is still a challenging and effective, even for adaptive methods. This confirms our data's high quality in exposing data leakage.
>
> [1] Learning to Plan Before Answering: Self-Teaching LLMs to Learn Abstract Plans for Problem Solving.
>
> ## About Input Noise Robustness from Reasoning
>
> We sincerely appreciate your concern about the effect of input noise. In fact, we incorporated a separate consideration of input noise in our evaluation. Our semi-fact OOD data is generated via two dynamic methods:
>
> - **Scenario-level Rephrasing**: Modifies a question's scenario and expression while preserving core knowledge.
> - **Attack-level Rephrasing**: introduces perturbations (e.g., noise) to simulate human-like errors (e.g., redundancy), following previous adversarial methods.
>
> As shown in Table 3 in the paper, Scenario-level OOD causes a larger performance drop than Attack-level OOD. For example:
>
> On AIME 500, Qwen2.5-Math-72B-IT scores 0.536 (original), 0.360 (Scenario-level OOD), and 0.500 (Attack-level OOD). Similarly, Qwen2.5-72B-IT achieves 0.432, 0.290, and 0.392, while LLaMA3.1-70B-IT scores 0.424, 0.244, and 0.392, respectively.
>
> The larger performance drop on Scenario-level OOD shows that robustness reflects the ability to handle core knowledge, not just resisting input noise. Thus, our semi-fact OOD data effectively evaluates reasoning ability. In the revision, we will further emphasize the distinction between our two levels of OOD data and provide a deeper analysis of their correlations to noise robustness and reasoning ability.
>
> ## About Comparison with Naturally Occurring OOD Data
>
> We deeply appreciate your suggestion to compare with natural OOD data. We used the newly published real exam AIME 2025 and compared model performance on AIME 2025 (natural OOD) against our constructed AIME 2024 OOD data.
>
> | Models              | AIME 2024 | AIME 2024 (OOD) | AIME 2025 |
> | ------------------- | --------- | --------------- | --------- |
> | deepseek-r1         | 0.800     | 0.750           | 0.667     |
> | s1.1-32B            | 0.533     | 0.517           | 0.500     |
> | deepseek-v3         | 0.367     | 0.333           | 0.333     |
> | Qwen2.5-Math-72B-IT | 0.267     | 0.217           | 0.233     |
> | Qwen2.5-72B-IT      | 0.200     | 0.184           | 0.133     |
> | LLAMA3.1-70B-IT     | 0.200     | 0.167           | 0.133     |
> | Llama-3.1-8B-IT     | 0.000     | 0.017           | 0.033     |
>
> Naturally occurring OOD (AIME 2025) caused more severe performance drops than our constructed OOD (e.g., for DeepSeek-R1: 0.800→0.667 vs. 0.800→0.750). While the degradation from our data is less severe, it is still close to that of natural OOD. This suggests our automatically generated data, while not as high-quality as human-created OOD, is still effective for identifying and mitigating data leakage.

---

> > ### Comment · Reviewer_upCj · 2025-08-05
> >
> > The rebuttal addresses all my concern. I will keep my score of acceptance and raise my confidence slightly. Thanks to the authors.

---

> > > ### Author Response · Authors · 2025-08-05
> > >
> > > Thank you very much for your recognition and valuable feedback! We sincerely appreciate your insightful suggestions, which greatly help us refine our work.

---

### Official Review · Reviewer_8g3p · 2025-07-02

**Ethics Flags:** Safety and security
**Rating:** 5
**Confidence:** 4

**Summary:**

The paper introduces a dynamic OOD evaluation method and dataset, developed using scenario-specific rephrase agents and attack agents operating at both character and sentence levels. The crafted OOD dataset is approximately four times larger than the combined sizes of the original AIME and GPQA datasets. It enables the assessment of reasoning models' generalization capabilities in OOD scenarios and facilitates the detection of data leakage. The proposed dynamic evaluation method and dataset are accessible via the provided hyperlink.

**Additional Feedback:**

Refer to Limitations Weaknesses

**Dataset Code Accessibility:**

Yes

**Dataset Code Comments:**

- The paper explicitly states in the Abstract that datasets and code are openly available, with sufficient instructions to faithfully reproduce the main experimental results.

- The benchmarking methodology is clearly described in Section 4.1 of the paper and in the repository's readme file.

**Ethical Comments:**

Although the paper does not involve sensitive data, the dynamic generation of test samples could, in theory, be exploited to reverse-engineer training data or expose vulnerabilities in deployed models if used maliciously.

**Ethical Considerations:**

Yes, there are ethics concerns that require attention by the authors

**Final Justification:**

The response has effectively addressed my concerns. Notably, the insightful fine-grained analysis of the six types of data leakage clearly highlights the limitations in LLM models' core reasoning capabilities. Consequently, I have revised the evaluation score from 4 to 5 and suggest that the authors incorporate the supplementary findings and analyses from the response into the paper for further expansion.

**Limitations Weaknesses:**

1. The paper should provide more evidence to demonstrate the reliability of human evaluation on dynamically generated OOD datasets. While human evaluation was conducted on AIME 2024 samples to verify question legitimacy, the process involved only three undergraduate students compensated at $2.50 per entry (Section 3.2), and the paper merely presents a simple alignment score. This raises concerns about annotation reliability, depth, and potential evaluation bias. To enhance credibility, the paper could consider increasing annotator numbers, implementing blind evaluation, and employing established metrics like the Kappa statistic.

2. The diversity of the crafted OOD dataset represents another significant weakness. Although the authors acknowledge that scenario-level semi-fact generation employs only a single-path rephrasing strategy (see Sec. 6)—limiting OOD diversity—this constraint may inadequately challenge models' ability to generalize across structurally varied contexts. Consequently, the significance of the paper's conclusions is restricted.

3. The paper lacks fine-grained analysis of data leakage phenomena. Detecting leakage in AIME 2024 through the proposed OOD dataset is an important finding. While the authors reveal leakage by comparing gaps between original and OOD datasets in AIME 2024 versus AIME 500, they fail to identify which specific portions of AIME 2024 were leaked. Fine-grained analysis could be achieved by comparing prediction probabilities between original and OOD datasets on a sample-wise basis.

4. Experimental results primarily emphasize performance on math and science reasoning tasks (Section 4.2.1). It provides limited analysis (Section 4.2.2) regarding framework effectiveness for evaluating non-mathematical reasoning models or how different architectural choices impact performance outside mathematical domains. The paper should offer a more balanced assessment of model strengths and weaknesses across diverse reasoning types.

**Strengths Contributions:**

- The paper introduces a novel method for generating OOD evaluation datasets to assess reasoning model robustness and detect data leakage in test sets.

-  The proposed approach and resulting corpus represent a valuable addition to existing LLM evaluation benchmarks.

- The paper is well-written and clearly structured. Figures and tables are effectively utilized and well-explained to support the corresponding claims.

---

> ### Author Rebuttal · Authors · 2025-07-31
>
> Thank you for your thoughtful and constructive feedback. We sincerely appreciate your recognition of our work and your valuable suggestions for improvement.
>
> ## About Reliability of Human Evaluation
>
> We appreciate your concerns regarding the reliability of human evaluation on our dynamically generated OOD datasets. To address this limitation, we have significantly expanded our human evaluation:
>
> We recruited three additional graduate students to join the original three undergraduate annotators, bringing the total to six annotators (three undergraduates and three graduates). To ensure evaluation fairness, annotators were not informed which instances were original versus OOD data. They were instructed to assess: (1) consistency between corresponding questions, (2) validity of question-answer pairs (i.e., the answer correctly addressed the question), and (3) logical equivalence scoring (0: completely non-equivalent, 0.5: partially equivalent, 1: fully equivalent) and similarity scoring (0: completely dissimilar, 0.5: partially similar, 1: nearly identical).
>
> Beyond AIME 2024, we conducted a comprehensive human evaluation on AIME-500, GPQA Diamond, and MMLU-570, with 30 samples annotated per dataset. We also report Fleiss' Kappa [1] values to measure annotation reliability:
>
> | Datasets     | Logical Equivalence Score | Question Similarity Score | Valid Score | Fleiss’ kappa |
> | ------------ | ------------------------- | ------------------------- | ----------- | ------------- |
> | AIME 2024    | 0.84                      | 0.36                      | 1.00        | 0.83          |
> | AIME-500     | 0.88                      | 0.38                      | 1.00        | 0.77          |
> | GPQA Diamond | 0.80                      | 0.49                      | 1.00        | 0.81          |
> | MMLU-570     | 0.94                      | 0.54                      | 1.00        | 0.90          |
>
> We complemented human evaluation with DeepSeek-R1 automated scoring using similar prompts, achieving consistent results that validate our human annotations.
>
> | Datasets     | Logical Equivalence Score | Question Similarity Score | Valid Score |
> | ------------ | ------------------------- | ------------------------- | ----------- |
> | AIME 2024    | 0.92                      | 0.43                      | 1.0         |
> | AIME-500     | 0.87                      | 0.48                      | 1.0         |
> | GPQA Diamond | 0.95                      | 0.55                      | 1.0         |
> | MMLU-570     | 0.97                      | 0.57                      | 1.0         |
>
> The perfect Valid Scores (1.00) across all datasets confirm that constructed questions and answers remain properly aligned. The high logical equivalence scores (≥0.80) demonstrate that our OOD data preserves the core logical structure while introducing minor irrelevant distractors.
>
> It is also worth noting that higher question similarity is not necessarily better, as the intention of constructing semi-fact OOD data was to preserve the core knowledge of the questions while modifying their expression and scenarios.
>
> [1] Measuring nominal scale agreement among many raters.
>
> ## About Analysis Beyond Mathematical Reasoning
>
> Thank you for your valuable suggestion. We will augment our paper with a detailed analysis of non-mathematical reasoning to better explore the model's comparative strengths and weaknesses.
>
> In addition to our current evaluation of AIME 2024, AIME-500, GPQA Diamond, and MMLU presented in the paper, we have expanded our evaluation to include diverse reasoning types:
>
> We sampled 30 instances each from LogiQA (logical reasoning) and ARC_Challenge (commonsense reasoning), plus 270 instances from BBH (covering 27 reasoning subtasks, including symbolic reasoning). Using our dynamic construction method, we generated corresponding Semi-fact OOD data.
>
> | Models              | LogiQA-30 (Original) | LogiQA-30 (OOD) | ARC_C-30 (Original) | ARC_C-30 (OOD) | BBH-270 (Original) | BBH-270 (OOD) |
> | ------------------- | -------------------- | --------------- | ------------------- | -------------- | ------------------ | ------------- |
> | deepseek-r1         | 0.833                | 0.867           | 1.000               | 0.984          | 0.856              | 0.815         |
> | s1.1-32B            | 0.800                | 0.517           | 0.967               | 0.967          | 0.737              | 0.741         |
> | deepseek-v3         | 0.767                | 0.633           | 0.967               | 0.933          | 0.811              | 0.767         |
> | Qwen2.5-Math-72B-IT | 0.700                | 0.567           | 0.900               | 0.833          | 0.804              | 0.811         |
> | Qwen2.5-72B-IT      | 0.700                | 0.517           | 0.933               | 0.900          | 0.752              | 0.700         |
> | LLAMA3.1-70B-IT     | 0.633                | 0.667           | 0.867               | 0.750          | 0.678              | 0.652         |
> | Llama-3.1-8B-IT     | 0.367                | 0.367           | 0.500               | 0.567          | 0.507              | 0.493         |
>
> Models show varying performance degradation patterns: most pronounced on LogiQA (logical reasoning), minimal on ARC_Challenge (commonsense reasoning), and moderate on BBH (multiple reasoning tasks). This suggests current models possess reasonable commonsense capabilities but struggle with complex logical reasoning, often relying on memorized patterns rather than genuine understanding.
>
> Qwen2.5-Math-72B-IT demonstrates superior performance and robustness on logical and symbolic reasoning tasks compared to the general Qwen2.5-72B-IT, while showing comparable performance on commonsense reasoning. This indicates that mathematical training enhances logical reasoning capabilities but provides limited benefits for commonsense reasoning.
>
> These comprehensive evaluations demonstrate the robustness and generalizability of our dynamic OOD construction.
>
> ## About Diversity of OOD Dataset
>
> We sincerely appreciate your concern about OOD diversity and provide the following clarifications:
>
> Our dynamic construction approach enables scenario modification from multiple perspectives while ensuring accuracy and usability, demonstrating scenario-level diversity. Furthermore,  as mentioned in the previous response, our method's applicability across various reasoning tasks demonstrates task-level diversity, as evidenced by our experiments across mathematical, scientific, logical, and commonsense reasoning domains. These comprehensive evaluations demonstrate the robustness and generalizability of ThinkBench in various domains and contexts.
>
> Our work tries to reveal that current LLMs suffer from data leakage and overfitting to superficial linguistic patterns while lacking deeper reasoning capabilities. To exclude confounding surface-level factors and truly evaluate models' core abilities for the future work, we deliberately focus on variations in language, scenarios, and expressions. Therefore, we explore the reasoning performance by introducing scenario modifications and linguistic noise.
>
> ## About Fine-grained Analysis of Data Leakage
>
> We appreciate your concern about fine-grained analysis of data leakage. To provide more granular insights into data leakage, we conducted a comprehensive error analysis:
>
> We categorized model errors into six types: Lack of Analysis (direct answers without reasoning), Hallucination (introducing non-existent conditions), Wrong Approach Selection (incorrect methodological choices), CoT Breakdown (reasoning chain failures), Numerical Error (computational mistakes), and Other.
>
> Using DeepSeek-R1 annotation, we analyzed error distributions on AIME-500:
>
>  **Table1：Error distributions on the Original data**
>
> | **Models**      | **Lack of Analysis** | **Hallucination** | **COT breakdown** | **numerical error** | **Wrong Approach Selection** | **Other** | **Total** |
> | --------------- | -------------------- | ----------------- | ----------------- | ------------------- | ---------------------------- | --------- | --------- |
> | o3-mini         | 0                    | 17                | 8                 | 7                   | 4                            | 3         | 39        |
> | Llama3.1-70B-IT | 19                   | 66                | 87                | 38                  | 33                           | 45        | 288       |
>
>   **Table2：Error distributions on the Scenario-level OOD data**
>
> | **Models**      | **Lack of Analysis** | **Hallucination** | **COT breakdown** | **numerical error** | **Wrong Approach Selection** | **Other** | **Total** |
> | --------------- | -------------------- | ----------------- | ----------------- | ------------------- | ---------------------------- | --------- | --------- |
> | o3-mini         | 0                    | 13                | 22                | 12                  | 8                            | 21        | 76        |
> | Llama3.1-70B-IT | 42                   | 83                | 104               | 53                  | 57                           | 39        | 378       |
>
> Key findings include:
>
> - **Llama3.1-70B-IT** shows a significant increase in "Lack of Analysis" errors on OOD data, compared to the original data (19→42), indicating the model becomes unfamiliar with scenario-shifted problems and skips detailed reasoning. This suggests a substantial data leakage issue in the original AIME-500 problem scenarios for Llama3.1-70B-IT.
> - Both models exhibit notable increases in "Wrong Approach Selection" and "COT Breakdown" errors on OOD data, compared to the original data. It suggests that memorized approaches and CoT solutions from familiar questions cannot be directly applied to unfamiliar scenarios.
>
> These patterns expose substantial data leakage in both original question's scenarios and expressions, confirming that models rely heavily on memorized solutions rather than genuine reasoning capabilities.

---

> > ### Comment · Reviewer_8g3p · 2025-08-07
> >
> > Thank you for your response. The response addressed my concerns, particularly the insightful fine-grained analysis of the six types of data leakage, which clearly highlights the limitations in LLM models' core reasoning capabilities.

---

> > > ### Author Response · Authors · 2025-08-07
> > >
> > > We sincerely appreciate your recognition and valuable feedback! Thank you for your insightful and thorough suggestions, which greatly help us strengthen our work.

---

> ### Author Response · Authors · 2025-08-01
>
> We sincerely thank the reviewer for raising the ethical consideration. We take the concern seriously and provide the detailed response below.
>
> We acknowledge the concern that there exists a small possibility that ThinkBench could theoretically be exploited maliciously. Specifically, the dynamic generation of OOD test samples could potentially be misused to: Reverse-engineer training data by using numerous generated samples as queries and analyzing model outputs to reconstruct sensitive information from the training set. This may expose vulnerabilities in deployed models through systematic probing with crafted OOD examples.
>
> To address this concern, we will release ThinkBench under a restrictive license that explicitly prohibits its use for reverse-engineering training data, membership inference attacks, or malicious probing of deployed systems. All users must agree to these terms before using ThinkBench.
>
> Furthermore, we will add a comprehensive discussion in the Limitations section in the revision that explicitly outlines these potential risks and emphasizes the importance of responsible usage.
>
> We believe that the responsible use of ThinkBench is crucial for ensuring safety. We sincerely appreciate the reviewer's guidance in ensuring our work contributes positively to the field while minimizing potential for misuse.

---

### Official Review · Reviewer_Vtef · 2025-07-02

**Rating:** 4
**Confidence:** 3

**Summary:**

This paper introduces ThinkBench, a dynamic evaluation benchmark designed to robustly assess the reasoning capabilities of large language models (LLMs) under out-of-distribution (OOD) conditions. The benchmark employs semi-factual data generation techniques at both scenario and attack levels to construct 2912 OOD samples from existing math and science datasets (AIME-500, AIME 2024, GPQA Diamond). ThinkBench evaluates both reasoning and non-reasoning models, mitigating data contamination and testing generalization beyond memorization. The authors benchmark 16 LLMs and 4 process reward models, showing widespread robustness gaps and performance decay on OOD data. The framework also supports test-time scaling analysis, highlighting its utility for studying LLM generalization under varied computational budgets.

**Additional Feedback:**

In line 125~126, the two attack-level data generation strategy is explained in the same way. It would be better to highlight the difference, or combine the two into one bullet.

Line 147~148, the string "“5XeflW1ZJc” is not syntactically valid, which differentiates the "CheckList" and "StressTest" method.

**Dataset Code Accessibility:**

Yes

**Dataset Code Comments:**

Both the code for data generation and the dataset are easily accessible and well-documented.

**Ethical Considerations:**

No, there are no or only very minor ethics concerns

**Final Justification:**

After considering the author rebuttal and discussion, I have increased my score from 3 to 4. The rebuttal effectively clarifies several key concerns, particularly the distinctions between scenario-level and attack-level OOD data generation and the efforts made to expand the evaluation beyond math/science domains. The authors also provide additional experiments on logical and commonsense reasoning tasks (LogiQA, ARC-Challenge, BBH), which strengthen the claim that the framework can generalize to broader reasoning challenges. Some concerns remain regarding the claim "Current LLMs suffer from data leakage and overfitting to superficial linguistic patterns while lacking robust internal reasoning capabilities", as per the experimental results with commonsense reasoning tasks, most of the models show minor performance degradation from original to OOD tasks, which is counter-intuitive to this overfitting claim.

**Limitations Weaknesses:**

- The scenario-level data generation seems to overlap with sentence-level, as both add irrelevant information regarding the core question without altering it.

- The benchmark focus on mathematical and scientific reasoning tasks only, while the goal is to robustly evaluate the LLM's reasoning capability. Future work could consider expanding the task domains.

- Although the semi-factual data generation strategy is novel, there are already existing work on dynamically generating reasoning problems for evaluation [1]. Which could also evaluate the OOD reasoning performance of LLMs.

- The out-of-distribution (OOD) data is constructed with scenario- or attack-level semi-factual rephrasing, but the core reasoning logic and final answer remain unchanged. This raises questions about whether such examples truly constitute OOD data, as they do not alter the underlying distribution of reasoning patterns or numerical relationships.

[1] Lin, Bill Yuchen, et al. "ZebraLogic: On the Scaling Limits of LLMs for Logical Reasoning." arXiv preprint arXiv:2502.01100 (2025).

**Strengths Contributions:**

- This paper is well written, and the study evaluates a wide range of state-of-the-art models..
- The proposed dynamic OOD evaluation framework and semi-factual data generation pipeline is novel.
- The error analyses are informative regarding model performance.
- The experimental results reveals some potential data leakage issues, especially for the AIME-500 dataset.

---

> ### Author Rebuttal · Authors · 2025-07-31
>
> We sincerely appreciate your insightful and constructive feedback, and we hope that our responses will clarify your concerns:
>
> ## About Overlap between Scenario-level and Attack-level OOD data generation
>
> We appreciate your concern about the overlap between our two data generation approaches. However, these methods serve distinct purposes and operate at different granularities:
>
> (1)  Scenario-level Rephrasing focuses on shifting the contextual scenario of a question while preserving the core knowledge and answer. In contrast, Attack-level Rephrasing modifies sentence structures (e.g., adding noise, redundancy, or ambiguity) following prior adversarial methods to mimic human-like linguistic variations. While both approaches adhere to the principle of not altering the core knowledge, they differ significantly in their focus—scenario transformation versus syntactic modification—and in their granularity of intervention.
>
> (2) Separating these approaches enables fine-grained analysis of how OOD constructions at different levels affect model performance. This separation helps clarify which specific dimensions of rephrasing cause performance gaps, providing valuable insights for understanding model robustness across multiple facets of linguistic variation.
>
> We will explicitly highlight these distinctions in the revision and justify the value of each approach for comprehensive analysis. Additionally, we will clarify the differences among  Attack-level OOD generation methods, such as CheckList and StressTest.
>
> ## About Limited task domain coverage
>
> We sincerely appreciate your suggestion to expand beyond mathematical and scientific reasoning tasks. To address this limitation, beyond our analyses of AIME 2024, AIME-500, GPQA Diamond, and MMLU presented in the paper, we have conducted additional experiments across diverse reasoning domains:
>
> - **LogiQA** ( We sample 30 instances and construct LogiQA-30): A representative logical reasoning dataset
> - **ARC_Challenge** (We sample 30 instances and construct ARC_C-30): A commonsense reasoning dataset
> - **BBH** (We sample 30 instances and construct BBH-270): 10 instances per category from 27 subtasks including symbolic reasoning
>
> Using our dynamic construction method, we generated corresponding Scenario-level and Attack-level Semi-fact OOD data and computed OOD and original performance scores, with the results as follows:
>
> | Models              | LogiQA-30 (Original) | LogiQA-30 (OOD) | ARC_C-30 (Original) | ARC_C-30 (OOD) | BBH-270 (Original) | BBH-270 (OOD) |
> | ------------------- | -------------------- | --------------- | ------------------- | -------------- | ------------------ | ------------- |
> | deepseek-r1         | 0.833                | 0.867           | 1.000               | 0.984          | 0.856              | 0.815         |
> | s1.1-32B            | 0.800                | 0.517           | 0.967               | 0.967          | 0.737              | 0.741         |
> | deepseek-v3         | 0.767                | 0.633           | 0.967               | 0.933          | 0.811              | 0.767         |
> | Qwen2.5-Math-72B-IT | 0.700                | 0.567           | 0.900               | 0.833          | 0.804              | 0.811         |
> | Qwen2.5-72B-IT      | 0.700                | 0.517           | 0.933               | 0.900          | 0.752              | 0.700         |
> | LLAMA3.1-70B-IT     | 0.633                | 0.667           | 0.867               | 0.750          | 0.678              | 0.652         |
> | Llama-3.1-8B-IT     | 0.367                | 0.367           | 0.500               | 0.567          | 0.507              | 0.493         |
>
> The results demonstrate that models generally exhibit performance degradation on OOD data across all three reasoning types, with the most pronounced decline on LogiQA (logical reasoning) and less significant drops on ARC_Challenge (commonsense reasoning). This suggests that current models possess a certain degree of commonsense reasoning and symbolic reasoning capabilities. However, in more complex logical reasoning tasks, models appear to rely on memorized answers, leading to poorer performance on OOD data. This indicates that they have not truly mastered sophisticated logical reasoning and suffer from severe data leakage issues.
>
> Additionally, we observe that Qwen2.5-72B-IT exhibits greater stability on OOD data from the ARC_Challenge compared to Qwen2.5-Math-72B-IT. However, it underperforms Qwen2.5-Math-72B-IT in terms of both performance and robustness on the BBH and LogiQA. This suggests that incorporating additional mathematical training data enhances capabilities in logical reasoning and symbolic reasoning tasks, while lacking the improvement in commonsense reasoning. This observation aligns with the phenomenon illustrated in Figure 6 in the paper regarding MMLU performance. This suggests commonsense reasoning correlates more strongly with knowledge-oriented benchmarks, while logical/symbolic reasoning ties closer to mathematical tasks.
>
> These results demonstrate the robustness and generalizability of our dynamic OOD construction method across multiple reasoning tasks. Furthermore, our findings confirm the existence of data leakage in these tasks, while our method can partially mitigate such leakage, providing a fairer and more comprehensive evaluation framework for existing models.
>
> ## About Comparison with existing dynamic generation work
>
> We deeply appreciate your concern about the existing dynamic generation work. Dynamic data construction methods, represented by ZebraLogic, employ a dynamically generated approach to evaluate logical ability in LLMs, which we also cited in our paper. Our work differs from ZebraLogic in its primary focus: ZebraLogic emphasizes the generation of puzzles with controllable and quantifiable complexity, whereas our method concentrates on the reformulation of existing problems.
>
> Our approach enables dynamic rewriting for countless tasks and scenarios, allowing the adaptation of any reasoning problem to generate semi-fact OOD data, thereby mitigating data leakage across diverse reasoning tasks. Furthermore, our framework provides Scenario-level and Attack-level dynamic rewriting, offering a multi-granular and comprehensive analysis of model reasoning capabilities.
>
> ## About Validity of OOD
>
> We appreciate your concern about whether our constructed data truly constitutes out-of-distribution data. Our dynamic data construction deliberately preserves underlying numerical relationships and reasoning logic.
>
> Current LLMs suffer from data leakage and overfitting to superficial linguistic patterns while lacking robust internal reasoning capabilities. Our objective is to eliminate confounding surface-level factors and help to rigorously evaluate models' core reasoning ability for the future work. The term "out-of-distribution" in our work specifically focuses on the variations in language, scenarios, and expressions—not to changes in the fundamental reasoning patterns or numerical relationships, which we intentionally preserve to ensure fair evaluation of reasoning abilities. We will further clarify and emphasize this focus in the revision to prevent misunderstanding.
>
> Furthermore, various and extensive experimental results demonstrate that our dynamic data construction can effectively mitigate the impact of data contamination.
>
> Thank you again for your valuable feedback, which helps us improve the clarity and comprehensiveness of our work.

---

> > ### Comment · Reviewer_Vtef · 2025-08-03
> >
> > Thanks for your response. My major concerns has been addressed and I have raised the score from 3 to 4.

---

> > > ### Author Response · Authors · 2025-08-05
> > >
> > > We sincerely appreciate your recognition and thorough review! Thank you for the constructive feedback that strengthens our work.

---

### Official Review · Reviewer_9n5N · 2025-07-03

**Rating:** 4
**Confidence:** 1

**Summary:**

​​ThinkBench introduces a novel framework for robustly evaluating reasoning capabilities in large language models (LLMs) by dynamically generating out-of-distribution (OOD) datasets. It addresses critical issues of data contamination and answer leakage through two core innovations: (1) a dynamic data generation method producing scenario-paraphrased and adversarially perturbed (stresstest, checklist, textbugger) variants of reasoning tasks, and (2) a unified benchmark with 2,912 OOD samples that enables direct comparison of 16 LLMs and 4 reasoning models (PRMs). Experiments reveal significant performance degradation (avg. 32% drop) and data leakage vulnerabilities in state-of-the-art models, demonstrating that ThinkBench effectively mitigates evaluation biases.

**Dataset Code Accessibility:**

Yes

**Ethical Considerations:**

No, there are no or only very minor ethics concerns

**Final Justification:**

After reviewing the authors' comprehensive rebuttal, I maintain my score of 4 (Borderline Accept). The authors have successfully addressed major concerns through expanded human evaluation across multiple datasets (AIME-500, GPQA Diamond, MMLU-570) with strong inter-rater reliability (Fleiss' Kappa 0.77-0.90), concrete failure case analysis showing how verifier agents filter problematic perturbations, and demonstration of broader applicability to logical reasoning (LogiQA), commonsense (ARC_Challenge), and symbolic tasks (BBH). However, the semi-factual causality framework remains primarily operational rather than formally grounded, and the benchmark scale (2,912 samples) is still limited compared to established benchmarks like MATH (12.5K). The work makes valuable contributions to addressing data contamination and evaluation robustness, but these remaining limitations prevent a higher score. The dynamic OOD generation method and systematic evaluation framework represent solid advances that warrant acceptance despite these constraints.

**Limitations Weaknesses:**

1. While the paper introduces dynamic scenario paraphrasing to create semantically consistent OOD data, it lacks comprehensive validation that these rephrasings reliably preserve logical equivalence across diverse domains. Although a human evaluation is conducted on AIME 2024, similar validation is absent for other tasks (e.g., GPQA or MMLU), and no automated consistency metrics (e.g., entailment scoring or semantic similarity) are reported.

2. The application of semi-factual causality is primarily conceptual and remains underformalized. The paper does not provide a rigorous framework for how such perturbations induce distributional shifts without altering reasoning chains. Furthermore, it lacks a systematic analysis of failure cases—such as when adversarial attacks (e.g., TextBugger) inadvertently disrupt the logical validity of the original problem.

3. The constructed benchmark, while methodologically novel, remains relatively small in scale—2,912 OOD samples derived from only three datasets (AIME-500, AIME 2024, GPQA Diamond). Compared to large-scale reasoning benchmarks like MATH (12.5K problems) or TheoremQA, the limited task diversity and sample size may constrain generalizability and statistical robustness of the findings.

**Strengths Contributions:**

​​1. This paper Introduces a novel scenario-paraphrasing (e.g., recontextualizing math problems into real-world scenarios) and adversarial perturbation framework (stresstest/checklist/textbugger) to create semantically consistent but distributionally shifted data. This addresses ​​data contamination​​ (e.g., GPT-4’s AIME performance drop) more effectively than static benchmarks

2. This paper surpasses perturbation tools (TextFlint, Checklist) via task-aware transformations preserving reasoning integrity.


3. This paper leverages semi-factual causality theory to ensure perturbations preserve logical equivalence while altering surface forms, enhancing robustness evaluation.

---

> ### Author Rebuttal · Authors · 2025-07-31
>
> Thank you for your recognition and constructive feedback, and we hope that our reply can solve your confusion.
>
> ## About Comprehensive Validation
>
> We sincerely appreciate the importance of comprehensive validation of data. To address this concern, we have expanded our human evaluation and automated assessments across multiple datasets.
>
> We recruited three additional graduate students to join the original three undergraduate annotators, bringing the total to six annotators (three undergraduates and three graduates). To ensure evaluation fairness, annotators were not informed which instances were original versus OOD data. They were instructed to assess: (1) consistency between corresponding questions, (2) validity of question-answer pairs (i.e., the answer correctly addressed the question), and (3) logical equivalence scoring (0: completely non-equivalent, 0.5: partially equivalent, 1: fully equivalent) and similarity scoring (0: completely dissimilar, 0.5: partially similar, 1: nearly identical).
>
> Beyond AIME 2024, we conducted a comprehensive human evaluation on AIME-500, GPQA Diamond, and MMLU-570, with 30 samples annotated per dataset. We also report Fleiss' Kappa [1] values to measure annotation reliability:
>
> | Datasets     | Logical Equivalence Score | Question Similarity Score | Valid Score | Fleiss’ kappa |
> | ------------ | ------------------------- | ------------------------- | ----------- | ------------- |
> | AIME 2024    | 0.84                      | 0.36                      | 1.00        | 0.83          |
> | AIME-500     | 0.88                      | 0.38                      | 1.00        | 0.77          |
> | GPQA Diamond | 0.80                      | 0.49                      | 1.00        | 0.81          |
> | MMLU-570     | 0.94                      | 0.54                      | 1.00        | 0.90          |
>
> We complemented human evaluation with DeepSeek-R1 automated scoring using similar prompts, achieving consistent results that validate our human annotations.
>
> | Datasets     | Logical Equivalence Score | Question Similarity Score | Valid Score |
> | ------------ | ------------------------- | ------------------------- | ----------- |
> | AIME 2024    | 0.92                      | 0.43                      | 1.0         |
> | AIME-500     | 0.87                      | 0.48                      | 1.0         |
> | GPQA Diamond | 0.95                      | 0.55                      | 1.0         |
> | MMLU-570     | 0.97                      | 0.57                      | 1.0         |
>
> The perfect Valid Scores (1.00) across all datasets confirm that constructed questions and answers remain properly aligned. The high logical equivalence scores (≥0.80) demonstrate that our OOD data preserves the core logical structure while introducing minor irrelevant distractors.
>
> It is also worth noting that higher question similarity is not necessarily better, as the intention of constructing semi-fact OOD data was to preserve the core knowledge of the questions while modifying their expression and scenarios.
>
> [1] Measuring nominal scale agreement among many raters.
>
> ## About Formalization of Semi-Factual Causality
>
> We appreciate the reviewer's concern about the conceptual nature of semi-factual causality in our work. Semi-factual reasoning is indeed a high-level concept drawn from prior causal theory and XAI literature [2,3]. In our paper, we adopted this concept as a practical methodology rather than proposing new causal theory. We agree that a more explicit formal definition would strengthen the presentation. In the revision, we will clarify our formal assumptions: specifically, that semi-factual perturbations change surface details of a problem without changing its solution or core reasoning. Our approach deliberately ensures that the correct answer remains the same for semi-factual variants. By building on established definitions, we focused on implementing semi-factual generation (scenario-level and attack-level) to create challenging OOD cases. We will make sure to emphasize in the paper that this approach is grounded in the literature and outline its theoretical basis more rigorously, to address the under-formalization concern.
>
> [2] Post-hoc Explanation Options for XAI in Deep Learning: The Insight Centre for Data Analytics Perspective.
>
> [3] On Generating Plausible Counterfactual  and Semi-Factual Explanations for Deep Learning.
>
> ## About Perturbations, Distributional Shifts, and Reasoning Chains
>
> We sincerely appreciate your concern of distributional shifts. We did not present a new formal framework for how our perturbations induce distribution shifts without altering reasoning chains. Instead, we operationalized semi-factual perturbations through a careful data generation pipeline. Notably, each perturbation was designed to preserve the original problem’s logical structure and solution path while yielding a novel surface form. To ensure this, we included Verifier Agents in our pipeline that automatically checks that all core reasoning aspects remain consistent and intact. Human and automated evaluation,  as mentioned in the previous response, found 100% solution alignment between original and semi-factual questions, indicating perturbations didn't alter reasoning chains or answers. We will clarify these implementation details and explicitly describe this verification step and our assumptions (e.g., defining "core reasoning chain" and how we preserve it) to provide a more rigorous underpinning for the semi-factual perturbation process.
>
> ## About Failure Case Analysis
>
> We appreciate your concern regarding the analysis of failure cases. We present a thorough investigation of failure cases encountered during our previous attempt.
>
> For instance, the original problem was: "There exist real numbers $x$ and $y$, both greater than 1, such that $\log_x(y^x) = \log_y(x^{4y}) = 10$. Find $xy$. " After rephrasing and modifying the scenario, it was incorrectly rewritten as: "In a magical garden, there are two unique types of flowers whose quantities are given by real numbers $x$ and $y$, where $a>1$ and $b>1$. They follow a special logarithmic relationship$\log_a(b^a) = \log_b(a^{4b}) = 10$. Find $xy$." This type of failure case arises from contextual variable confusion, which causes the evaluation model to be uncertain about what to compute, leading to incorrect answers. As a result, it fails to fairly assess the model's reasoning capability. We will include a systematic analysis of such failure cases in the revision.
>
> Notably, ThinkBench is able to effectively filter such failure cases through iterative verifier's validation, ensuring contextual coherence and consistency. The high validity scores (1.00) across all evaluated datasets demonstrate that our framework successfully maintains high quality and logical validity while introducing expression and scenario distributional shifts.
>
> ## About Scale and Generalizability
>
> We deeply appreciate the concern regarding benchmark scale and diversity.  Currently, most evaluations of large models are conducted using benchmark datasets such as AIME 2024, Math 500, and GPQA Diamond. Therefore, our initial focus was on mathematical and scientific reasoning domains, and selected AIME 2024, AIME-500, GPQA Diamond, and MMLU as the original datasets to build and analyze OOD data. Using ThinkBench, it is possible to dynamically generate more OOD data tailored to various reasoning tasks, exposing and mitigating data leakage.
>
> While our initial focus was on mathematical and scientific reasoning domains, we have expanded our evaluation to demonstrate broader applicability and robustness: We sampled 30 instances each from LogiQA (logical reasoning) and ARC_Challenge (commonsense reasoning), plus 270 instances from BBH (27 reasoning subtasks, including symbolic reasoning).
>
> Using our dynamic construction method, we generated corresponding Semi-fact OOD data, with the performance as follows:
>
> | Models              | LogiQA-30 (Original) | LogiQA-30 (OOD) | ARC_C-30 (Original) | ARC_C-30 (OOD) | BBH-270 (Original) | BBH-270 (OOD) |
> | ------------------- | -------------------- | --------------- | ------------------- | -------------- | ------------------ | ------------- |
> | deepseek-r1         | 0.833                | 0.867           | 1.000               | 0.984          | 0.856              | 0.815         |
> | s1.1-32B            | 0.800                | 0.517           | 0.967               | 0.967          | 0.737              | 0.741         |
> | deepseek-v3         | 0.767                | 0.633           | 0.967               | 0.933          | 0.811              | 0.767         |
> | Qwen2.5-Math-72B-IT | 0.700                | 0.567           | 0.900               | 0.833          | 0.804              | 0.811         |
> | Qwen2.5-72B-IT      | 0.700                | 0.517           | 0.933               | 0.900          | 0.752              | 0.700         |
> | LLAMA3.1-70B-IT     | 0.633                | 0.667           | 0.867               | 0.750          | 0.678              | 0.652         |
> | Llama-3.1-8B-IT     | 0.367                | 0.367           | 0.500               | 0.567          | 0.507              | 0.493         |
>
> Performance degradation varied: highest on LogiQA (logical), minimal on ARC_Challenge (commonsense). This suggests current models possess reasonable commonsense but struggle with complex logical reasoning, often relying on memorized patterns rather than genuine understanding.
>
> Compared to Qwen2.5-72B-IT, Qwen2.5-Math-72B-IT shows better performance and robustness on logical/symbolic tasks but comparable commonsense ability. This suggests more math training enhances logical reasoning but provides limited improvement in commonsense reasoning.
>
> These results confirm our dynamic OOD method is robust and generalizable. Our findings reveal data leakage in these tasks, and our method provides a fairer and more comprehensive evaluation framework.

---

> > ### Comment · Reviewer_9n5N · 2025-08-05
> >
> > Thank you for your thorough and thoughtful rebuttal. I appreciate the substantial effort you've made to address my concerns.
> >
> >
> > **Remaining concerns:**
> > 1. While you acknowledge semi-factual causality is adopted as a practical methodology, a more formal theoretical framework would strengthen the contribution
> > 2. The benchmark scale (2,912 samples) remains limited compared to MATH (12.5K) or TheoremQA
> >
> > **Suggestions for camera-ready:**
> > 1. Include the expanded validation results (human evaluation table) in the main paper or appendix
> > 2. Add a dedicated subsection on failure case analysis with the examples you provided
> > 3. Consider discussing future work on scaling the benchmark using your dynamic generation method
> >
> > Your work makes important contributions to addressing data contamination in LLM evaluation. The dynamic OOD generation approach and systematic evaluation framework are valuable additions to the field. I look forward to seeing the final version with these improvements incorporated.

---

> > ### Author Response · Authors · 2025-08-06
> >
> > Thank you very much for your constructive feedback. We greatly appreciate your recognition of our work's contributions. In the camera-ready version, we promise to incorporate our expanded validation results (human evaluation table), and add a dedicated subsection on failure case analysis with detailed examples. Furthermore, we will discuss future work on scaling the benchmark using our dynamic generation method. We will strengthen the theoretical foundation of our semi-factual causality with more formal assumptions and definitions.
> >
> > Thank you once again for your insightful and valuable suggestions, which greatly help us strengthen our work. We sincerely hope that our work can make a better impact on the research field!

---

> > > ### Comment · Reviewer_9n5N · 2025-08-08
> > >
> > > Thank you for the detailed response. I have no further concerns and appreciate your thorough engagement with the review feedback.

---

### Comment · Area_Chair_ryU4 · 2025-08-08

Dear Reviewers,

First, I want to appreciate your help in contributing NeurIPS review process. According to this year's policy, we may flag reviewers without any interaction with the authors -- so please let the authors and us know if they addressed your concerns.

Appreciate your help again!

---

### Note · Authors · 2025-08-13

We sincerely thank all reviewers for their constructive feedback and are pleased that our responses successfully addressed their concerns.

During the rebuttal, we engaged extensively with reviewers to address their concerns through comprehensive discussions and experimental additions:

**Comprehensive Validation (Reviewers 9n5N, 8g3p):** Reviewers raised concern about validation. To demonstrate data's reliability, we expanded comprehensive human evaluation across AIME-500, GPQA Diamond, MMLU-570 with Fleiss' Kappa values and added automated validation. Perfect valid scores (1.00) and high logical equivalence scores (≥0.80) proved our data's reliability, addressing reviewers' validation concern.

**Generalizability (Reviewers 9n5N, Vtef, 8g3p, upCj):** Reviewers posed the concern about limitation to mathematical/scientific reasoning. To validate ThinkBench's generalizability and robustness, we added experiments on 330 samples covering LogiQA, ARC_Challenge, and BBH across diverse reasoning types. Consistent OOD degradation across diverse reasoning types proved broad generalizability, addressing reviewers' concern.

**Detailed Analysis of Data Leakage (Reviewers 8g3p, upCj):** Reviewers proposed the concern about detailed data leakage analysis. To provide finer-grained analysis, we added a six-category error classification to investigate data leakage. Clear error distributions revealed distinct leakage types, directly addressing the reviewers' concern.

**Adaptive Baseline and Naturally Occurring OOD Analysis (Reviewer upCj):** Reviewer posed the concern about the comparison with adaptive baseline and natural OOD dataset. To validate practical effectiveness, we added prompt-based planning baseline comparison and AIME 2025 natural OOD comparison. Results show that our data is high quality and effective for identifying and mitigating data leakage, addressing reviewer's concern.

**Ethical Safety (Ethics Reviewers):** Reviewers raised the concern about the potential misuse of ThinkBench. We addressed potential misuse concern through comprehensive safety framework discussion and restrictive licensing implementation.

The comprehensive experiments and discussions demonstrate ThinkBench's robustness and effectiveness in identifying and mitigating data leakage. We sincerely thank all reviewers for their insightful and valuable suggestions, which greatly helped us strengthen our work. We sincerely hope that ThinkBench can make a better impact on the research field!

---

### Decision · Program_Chairs · 2025-09-18

**Decision:**

Accept (poster)

**Comment:**

This work proposes a dynamic OOD evaluation recipe and a small public benchmark (~3k items) to test LLM reasoning under paraphrase/attack shifts. It runs a large number of LLMs and PRMs for it to show the OOD drops and leakages.

Good fit to DB track (evluation, benchmark, and reproducibility).

Mostly positive after discussion. It ended at two accept (5) and two at borderline accept (4). Main pre‑rebuttal concerns:
(i) validity of “semi‑factual” rewrites (do they preserve the solution?),
(ii) STEM‑heavy scope and small scale,
(iii) limited analysis of leakage types,
(iv) what “OOD” really means here vs simple noise.

I personally feel (i) is a bit more pressing, (ii) is fine as it is just one benchmark paper not expected to cover everything. (iii) and (iv) are not critical.

The rebuttal is extensive and the author add more validation, baselines, remidies for ethic reviews. Overall,  clear, useful benchmark + methodology to reduce contamination effects; strong community interest; solid engagement with reviewers. This is sufficient for acceptance.